# Plug and Play Language Models: a Simple Approach to Controlled Text Generation

**Sumanth Dathathri** [*]
CMS, Caltech

**Andrea Madotto** [*]
HKUST

**Janice Lan**
Uber AI

**Jane Hung**
Uber AI

**Eric Frank**
Uber AI

**Piero Molino**
Uber AI

**Jason Yosinski** [†]
Uber AI

**Rosanne Liu** [†]
Uber AI

```
dathathris@gmail.com, amadotto@connect.ust.hk
{janlan, jane.hung, mysterefrank, piero, yosinski, rosanne}@uber.com
```

## Abstract

Large transformer-based language models (LMs) trained on huge text corpora have shown unparalleled generation capabilities. However, controlling attributes of the generated language (e.g. switching topic or sentiment) is difficult without modifying the model architecture or fine-tuning on attribute-specific data and entailing the significant cost of retraining. We propose a simple alternative: the Plug and Play Language Model (PPLM) for controllable language generation, which combines a pretrained LM with one or more simple attribute classifiers that guide text generation without any further training of the LM. In the canonical scenario we present, the attribute models are simple classifiers consisting of a user-specified bag of words or a single learned layer with 100,000 times fewer parameters than the LM. Sampling entails a forward and backward pass in which gradients from the attribute model push the LM's hidden activations and thus guide the generation. Model samples demonstrate control over a range of topics and sentiment styles, and extensive automated and human annotated evaluations show attribute alignment and fluency. PPLMs are flexible in that any combination of differentiable attribute models may be used to steer text generation, which will allow for diverse and creative applications beyond the examples given in this paper.

## 1 Introduction

The Transformer architecture (Vaswani et al., 2017) has enabled large-scale language models (LMs) trained on a huge amount of data (Radford et al., 2019; Dai et al., 2019b; Radford et al., 2018b) to greatly improve the state-of-the-art on natural language processing tasks. These models are used to extract contextualized word embeddings for transfer learning purposes (Devlin et al., 2019) and as natural language generators. The latter can leverage large amounts of unannotated data and a simple log-likelihood training objective. However, once such models are trained, controlling attributes of generated text becomes difficult without modifying the model architecture to allow for extra input attributes or fine-tuning with attribute-specific data (Keskar et al., 2019; Ziegler et al., 2019).

---

[*]Work done during internship at Uber AI

[†]Co-senior authors

• Summary of contributions: SD, RL & JY conceptualized PPLMs and led the manuscript writing. SD led the project, implemented the PPLM, set up and ran all modeling experiments, engineered how to obtain workable gradients via the weighted embedding approach, and made the model work. AM helped with preparing datasets for discriminator training, automated evaluation, running experiments, and writing the manuscript. SD, RL & AM ran the external baselines. RL & JL built and oversaw the human evaluation pipeline and computed the statistics. JH ran the story generation with skeleton prefixes. EF assisted with detoxification experiments. PM led efforts to migrate to the new pytorch transformer, helped with code release. JY helped with the annotation pipeline, finding bugs, navigating model and experimental directions, engineering workable gradients, and posing the model mathematically. RL implemented preliminary experiments and multi-attribute control, and cleaned and coordinated release of the code. RL & JY oversaw the project.

Table 1: The PPLM employs a pre-trained language model (LM) without any changes to the model parameters and can generate text with controlled attributes such as topic and sentiment. We demonstrate control with two tiny and easy to construct attribute models: a bag of words (BoW) related to a topic and a linear discriminator trained on top of LM latent representations to control sentiment. The underlined prefix is what the LM is conditioned on to generate a passage of text (e.g. The potato). The controlled attributes are colored and bracketed (e.g. **[Science]** ), and words in the BoW that are directly optimized for are highlighted brightly (e.g. research). The softer highlights correspond to words related to the attribute, but not directly optimized for during the control process (e.g. health).

| |
|---|
| **[–]** The potato and cauliflower are both in season to make combo breads, mounds, or pads. For an added challenge, try some garlic mashed potatoes. |
| **[Negative]** The potato is a pretty bad idea. It can make you fat, it can cause you to have a terrible immune system, and it can even kill you... |
| **[Positive]** The potato chip recipe you asked for! We love making these, and I've been doing so for years. I've always had a hard time keeping a recipe secret. I think it's the way our kids love to eat them – so many little ones. |
| **[Science]** The potato was once thought to have no health problems and has been promoted as a nutritious food source since the mid-1800s, but recent reports indicate that it has many harmful health issues. In fact, researchers from Johns Hopkins University... |
| **[Politics]** **[Positive]** To conclude this series of articles, I will present three of the most popular and influential works on this topic. The first article deals with the role of women's political participation in building a political system that is representative of the will of the people. |
| **[Politics]** **[Negative]** To conclude, the most significant and lasting damage from the economic crisis in 2008 was that many governments, including those in the political center, lost power for the first time in modern history. |

Controllable generation entails modeling $p(x|a)$, where $a$ is some desired controllable attribute(s) and $x$ the generated sample. However, generative models only learn $p(x)$. In computer vision, Plug & Play Generative Networks (PPGN) from Nguyen et al. (2017) developed a mechanism for generating images with different attributes by plugging a discriminator (attribute model) $p(a|x)$ together with a base generative model $p(x)$ and sampling from the resulting $p(x|a) \propto p(a|x)p(x)$, effectively creating a conditional generative model on the fly from any supplied attribute model. In a similar manner, we propose the Plug and Play Language Model (PPLM) for conditional language generation that combines one or more simple attribute models $p(a|x)$—either in the form of a bag-of-words (BoW) or single layer classifiers—with a pre-trained, unconditional language model $p(x)$. We sample from the resulting combined model by following gradients in the latent representation space in a manner inspired by the approximate Metropolis-adjusted Langevin (MALA) (Roberts et al., 1996; Roberts & Rosenthal, 1998) sampler deployed in Nguyen et al. (2017).

Optimization is performed *ex post facto* in the activation space, therefore *no re-training or fine-tuning is needed*. Control is fine-grained, with a strength parameter determining how strong the attribute influence should be; a strength of $0$ fully recovers the original model $p(x)$. This design allows vast flexibility: users can combine a state-of-the-art generative model, which may be large and difficult to train, with any number of attribute controllers. Attribute models may be easier to train or untrained (in the case of BoW models), and multiple controllers may be combined flexibly during inference. In this paper, we demonstrate the PPLM approach using a GPT-2 345M model (Radford et al., 2019) as the general-purpose LM $p(x)$, but the method applies in any representation space from any transformer-based text generator and allows combination with any attribute model $p(a|x)$.

We demonstrate controlled generation with a number of attribute controllers, assembled and combined during generation, each with a different strength, acting as a set of "control knobs" that tune generation towards the desired attribute (see examples in Table 1). Code for the experiments is available at: `https://github.com/uber-research/PPLM`. Our key contributions are:

- We introduce the Plug and Play LM for controlled language generation, discuss its relation to existing work, and how sampling from a PPLM works (Sections 2 and 3).

- We demonstrate controlling of text generation on a range of attributes, including 7 topics each defined using a bag of words, and 1 simple discriminator on sentiments. We quantify effectiveness using both automated evaluation (separately trained perplexity and sentiment

models) as well as human evaluation (for attribute relevance and fluency). All evaluations point toward the ability of PPLMs to generate attribute controlled, fluent text (Section 4).

- We compare PPLM with CTRL (Keskar et al., 2019) and GPT-2 finetuned for positivty (Ziegler et al., 2019). Our method, without any LM training, is on par and often outperforms the baselines on attribute relevance and fluency (Section 4.2, and Section 4.3).

- We show that the PPLM approach can be used to detoxify instances where generation of toxic content is likely by following the negative gradient of a model trained to detect toxicity (Section 4.4). We also show how PPLM can be used for structurally constrained story writing (Section 4.5).

## 2 RELATED WORK

**Controlled generation**   Current methods for controlled text generation involve either fine-tuning existing models with Reinforcement Learning (RL) (Ziegler et al., 2019), training Generative Adversarial Networks (Yu et al., 2017), or training conditional generative models (Kikuchi et al., 2016; Ficler & Goldberg, 2017).   Different from our approach, these methodologies are not plug and play, since the entire model needs to be separately fine-tuned for each specific attribute. Keskar et al. (2019) train a large language model with over 50 different control codes. The results are high quality because they train exactly to maximize $p(x|a)$, but this comes at the expense of fixing control codes upfront and of training a very large model (1.6B parameters). Our method does not require retraining any conditional generative model, and both the language model and the conditional model can be flexibly assembled. Table 2 gives a comparison of recent approaches to language modeling tuned for specific attributes. In another interesting but tangential piece of work, Subramani et al. (2019) recently showed that a pre-trained language model can be steered to recover arbitrary sentences. In earlier works Gu et al. (2016; 2017); Chen et al. (2018) explored the idea of using a small neural network to steer an LM.

**Noisy Channel Modeling**   Yu et al. (2016), and more recently Yu et al. (2019); Yee et al. (2019); Ng et al. (2019), leveraged the Shannon Noisy Channel Theory (Shannon, 1948) for improving sequence-to-sequence modeling. Their approach translates a source language sentence $y$ into a target language sentence $x$ by first sampling from a forward model proposal distribution $p_{\text{forward}}(x|y)$ and then reranking samples based on probabilities given by $p_{\text{backward}}(x|y) \propto p(x)p(y|x)$. PPLM scores samples using the same basic equation, but as we have no forward or proposal model $p_{\text{forward}}(x|a)$, we rely on the latent space updates, similar to Nguyen et al. (2017). As a baseline, we consider using $p(x)$ as a "forward model" and then reranking, which we will see works moderately well in some scenarios and poorly in others (see Tables 4 and 6).

**Weighted decoding**   Holtzman et al. (2018); Ghazvininejad et al. (2017) consider controlled language generation – the former with discriminators, and the latter with a bag of words – where the decoding procedure is modified to consider the scoring function used for decoding. See et al. (2019) note that control with weighted decoding (WD) is difficult and often leads to sacrificing fluency and coherence. Further, Ghazvininejad et al. (2017) strongly relies on sampling from a set of keywords on a specific topic and it does not allow to bias generation towards a topic in a manner that does not necessary include a set of keywords. Similarly, Baheti et al. (2018) proposed a decoding strategy for generating interesting responses in dialogue systems, using bags of words and word embeddings. Sophisticated sampling methods (Metropolis et al., 1953) can be used to constrain the model generation to certain keywords and topics. We evaluate WD as a baseline.

**Text Style Transfer**   Outside of language modeling, the text style transfer studies a related task. Shen et al. (2017); Hu et al. (2017) train variational auto-encoders for style transfer that rely on learning disentangled latent representations for style and content. Li et al. (2018) demonstrate the efficacy of a simple approach based on replacing attribute related n-grams with n-grams corresponding to the desired attribute based on a conditional generative model. A key difference between the above and our approach is that we use an offline discriminator and perform optimization based on this discriminator, which as suggested by Elazar & Goldberg (2018) may outperform adversarial training approaches. More recently, Lample et al. (2019) adapt an approach from unsupervised language translation to style transfer, where a denoised auto-encoder is trained with an objective

Table 2: Comparison of the different models and distributions. All models in this table are useful in different scenarios. The particular advantage of PPLM is that very small, custom attribute models, $p(a|x)$, may be combined with powerful, general pre-trained language models, $p(x)$, to create cheap but still powerful conditional generative models, $p(x|a)$.

| Model type | Form of model | Samples | Example models and number of trainable params |
|---|---|---|---|
| Language Model | $p(x)$ | Uncond. | GPT-2 medium: 345M (Radford et al., 2019) |
| Fine-tuned Language Model | $p(x)$ | Uncond. | Fine-tuned GPT-2 medium: 345M (Ziegler et al., 2019) |
| Conditional Language Model | $p(x|a)$ | Cond. | CTRL: 1.6B (Keskar et al., 2019) |
| Plug and Play Language Model (PPLM) | $p(x|a) \propto p(x)p(a|x)$ | Cond. | PPLM-BoW: 0 (curated word list) PPLM-Discrim: $\sim$ 1K/attribute (not counting pretrained $p(x)$) |

consisting of a weighted combination of a re-construction loss and a back-translation loss. While the above approaches have shown impressive success on style transfer tasks, the main focus is not controlled language generation, and further, the methods are not *plug and play*.

## 3 PLUG AND PLAY LANGUAGE MODELS

### 3.1 LANGUAGE MODELING WITH TRANSFORMERS

Given a sequence of tokens $X = \{x_0, \cdots, x_n\}$, LMs are trained to compute the unconditional probability of the sequence $p(X)$. This probability can be rewritten in terms of product of conditional probabilities by recursively applying the chain-rule (Manning et al., 1999; Bengio et al., 2003) as:

$$p(X) = \prod_{i=1}^{n} p(x_i|x_0, \cdots, x_{i-1}) \tag{1}$$

In this paper, we use a transformer (Vaswani et al., 2017) to model the distribution of natural language. To present our approach clearly, we first briefly summarize the transformer using recurrent notation. Let us define the history matrix $H_t$ to consist of the key-value pairs from the past i.e $H_t = [(K_t^{(1)}, V_t^{(1)}), \cdots, (K_t^{(l)}, V_t^{(l)})]$, where $(K_t^{(i)}, V_t^{(i)})$ corresponds to the key-value pairs from the $i$-th layer generated at all time-steps from 0 to $t$. Efficient implementations of the transformer (Wolf et al., 2019) use the cached $H_t$ to generate $x_{t+1}$, given $x_t$. This recurrent interpretation of a transformer can be summarized as:

$$o_{t+1}, H_{t+1} = \text{LM}(x_t, H_t), \tag{2}$$

and then $x_{t+1}$ is sampled as $x_{t+1} \sim p_{t+1} = \text{Softmax}(Wo_{t+1})$, where $W$ is a linear transformation that maps the logit vector $o_{t+1}$ to a vector of vocabulary size. This allows for efficient language generation without repeated forward passes corresponding to the prior conditioning text $x_0, \ldots, x_{t-1}$.

### 3.2 STEERING GENERATION: ASCENDING $\log p(a|x)$

In order to control the output of the language model, at every generation step $t$, we shift the history $H_t$ in the direction of the sum of two gradients: one toward higher log-likelihood (LL) of the attribute $a$ under the conditional attribute model $p(a|x)$ and one toward higher LL of the unmodified language model $p(x)$. Combining these factors with a variable multiplier provides us with a controllable "knob" to guide generation in a given direction with a specified strength. The updates are restricted to $H_t$ and not the other model activations because future predictions depend on the past only via $H_t$ (note that $H_t$ is composed of all transformer key and value pairs generated up to time $t$). Taking steps in $H_t$ space leads to gradual changes to model activations — which may be thought of as gradual reinterpretations of the past — that guide future generation in the desired direction.

Let $\Delta H_t$ be the update to $H_t$, such that generation with $(H_t + \Delta H_t)$ shifts the distribution of the generated text such that it is more likely to possess the desired attribute. $\Delta H_t$ is initialized

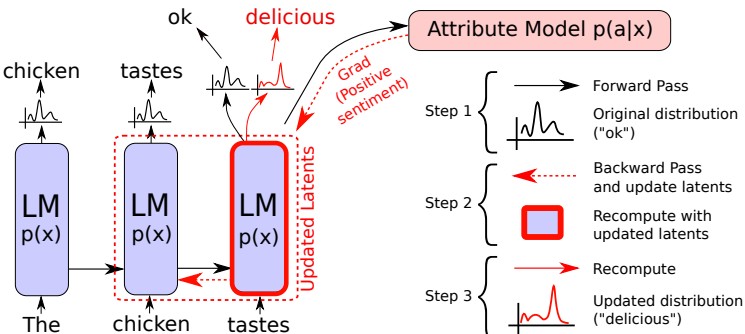

Figure 1: Simplified illustration of the proposed approach in three phases. In Step 1, a forward pass is performed through the language model to compute the likelihood of a desired attribute using an attribute model that predicts $p(a|x)$. In Step 2, a backward pass updates the internal latent representations of the LM, using gradients from the attribute model, to increase the likelihood of the passage having the desired attribute. In Step 3, a new distribution over the vocabulary $(\widetilde{p}_{t+1})$ is generated from the updated latents $(\widetilde{H}_t)$ and the current token $x_t$. The next token is then sampled from the updated distribution. This process of updating the latents is repeated at each time-step, leading to a gradual transition towards the desired attribute. For computational efficiency, one may choose to modify only the latents within some window of the recent past, depicted as the dotted-red region.

at zero and updated with gradients from an attribute model that measures the extent to which the generated text possesses the desired attribute (e.g. positivity). We rewrite the attribute model $p(a|x)$ as $p(a|H_t + \Delta H_t)$ and then make gradient based updates to $\Delta H_t$ as follows:

$$\Delta H_t \leftarrow \Delta H_t + \alpha \frac{\nabla_{\Delta H_t} \log p(a|H_t + \Delta H_t)}{\|\nabla_{\Delta H_t} \log p(a|H_t + \Delta H_t)\|^\gamma} \tag{3}$$

where $\alpha$ is the step size, $\gamma$ is the scaling coefficient for the normalization term.[1] This update step can be repeated $m$ times; in practice we use 3 to 10. Subsequently, a forward pass through the LM with the updated key-value pairs is performed to obtain the updated logits $\widetilde{o}_{t+1}$ as $\widetilde{o}_{t+1}, H_{t+1} = $ LM$(x_t, \widetilde{H}_t)$, where $\widetilde{H}_t = H_t + \Delta H_t$. The perturbed $\widetilde{o}_{t+1}$ is then used to generate a new distribution $\widetilde{p}_{t+1}$ as in Equation 2.

## 3.3 Ensuring fluency: ascending $\log p(x)$

The approach described in the previous section is able to generate text tuned for a particular discriminator, but left unchecked it will quickly result in unrealistic adversarial or fooling examples (Szegedy et al., 2013; Nguyen et al., 2015) as the text moves into low probability regions. To combat this, we use the unconditional language model in two ways that ensure the fluency is maintained at or near the level of the unconditional language model (here GPT-2).

**Kullback–Leibler (KL) Divergence** We update $\Delta H_t$ to minimize the KL divergence between the output distribution of the modified and unmodified language models in addition to the step above. In practice, this is accomplished by adding the quantities together before taking a gradient, though it can be visualized as two separate steps as in Figure 2. We scale the KL coefficient by a scalar $\lambda_{KL}$, and in practice, setting this hyperparameter to 0.01 works well in general across tasks.

**Post-norm Geometric Mean Fusion** In addition to minimizing KL divergence, which affects the past via $\Delta H_t$, we perform *post-norm fusion* similarly to Stahlberg et al. (2018). This does not directly affect $\Delta H_t$; rather, it just serves to constantly tie the generated text to the unconditional $p(x)$ LM distribution. We accomplish this by sampling from $x_{t+1} \sim \frac{1}{\beta} \left( \widetilde{p}_{t+1}^{\gamma_{gm}} p_{t+1}^{1-\gamma_{gm}} \right)$, where $p_{t+1}$ and $\widetilde{p}_{t+1}$ are the unmodified and modified output distributions, respectively, and $\beta$ is a normalizing factor such that it forms a valid distribution. As $\gamma_{gm} \to 1$ this converges to the distribution from the updated LM, and as $\gamma_{gm} \to 0$ it converges to the unconditional LM distribution. We find that in practice values for $\gamma_{gm}$ in the range $0.8 - 0.95$ work well.

---

[1] One normalization term is computed for each layer of the transformer.

Figure 2: An oversimplified view into why steps that maximize both $\log p(a|x)$ and $\log p(x)$ are needed. The sentence under consideration is shown as a black dot, which is first pushed in the direction of maximizing $\log p(a|x)$ and then in the direction of maximizing $\log p(x)$. In practice we use a single step and simply add the log probabilities; we take steps in continuous space of hidden representations $H$ rather than in the discrete $x$ (byte pair) space, and rather than resampling the entire sentence each step, we take one step in $H$ space per byte-pair sample.

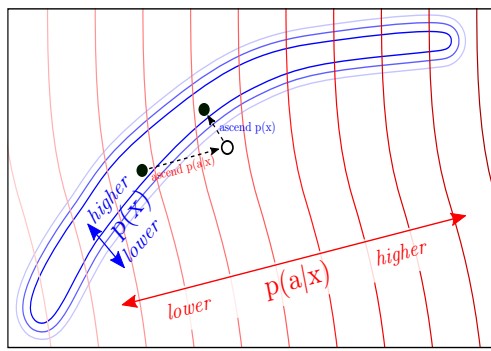

### 3.4 Sampling and Ranking

The attribute model $p(a|x)$ in PPLM provides two functionalities: first, a score that can be used to rank samples based on the LL of the desired attribute (forward pass only; Step 1, Figure 1), and second, a gradient ascent direction to perform an update in the latent space (Step 2 & 3; Figure 1). The former can be used to generate $r$ samples and rank them to choose the best one. This can serve as an additional method for attribute control in addition to sampling with updated latents. Further, to avoid the problem of repetitive, low quality text (Holtzman et al., 2018), we compute the mean over the Dist-1, Dist-2 and Dist-3 scores (for the generated passage), which is an indicator of repetitiveness (Li et al., 2015), and then discard samples with a mean score below a threshold $\tau$.

## 4 Experiments, Results, and Evaluation

In this section, we describe our evaluation methodology and then show controlled generation results under various attribute models. We also show use cases of PPLM in language detoxification and in controlled story telling. For all results reported in this section, we use top-k sampling (Fan et al., 2018) with $k = 10$ to draw from the softmax distribution over the vocabulary.

### 4.1 Evaluation methods and ablation study

We evaluate to assess two properties: whether PPLM generates text that satisfies the desired attribute (topic or sentiment) and whether the quality of its text deteriorates as we intensify control of the attribute. Note we can always turn the control knob down to zero to disable control of attributes and reach the fluency of the original model. If desired, a user can tune the knobs at inference until a chosen tradeoff between attribute strength and fluency is reached. We evaluate using both automated methods and human annotators:

**Automated Eval.** Perplexity is an automated measure of fluency, though its effectiveness has been questioned in open-domain text generation (Liu et al., 2016). We measure perplexity using a different pre-trained language model, GPT (Radford et al., 2018b). The diversity of text in the passages is measured using the number of distinct n-grams (normalized by the length of text) as in Li et al. (2015). We report Dist-1, Dist-2, and Dist-3 scores for the distinct 1-2-3-grams (measured across all samples generated for a given attribute control task, e.g. a specific topic for topic control). Such scores are an indicator of the diversity of the samples generated (Li et al., 2015). We also use external sentiment classifiers for sentiment evaluation.

**Human Eval.** We consider two types of human annotation: fluency and A/B testing on attribute relevance. Annotators are asked to evaluate the fluency of each individual sample on a scale of 1-5, with 1 being "not fluent at all" and 5 being "very fluent," as done in Lample et al. (2019). In the A/B testing for attribute relevance, we consider all combinatorial pairs of all four variants: B, BR, BC, and BCR (6 combinations). We then ask annotators to rank the pair on the desired attribute (e.g. topic relevance, sentiment strength), while allowing "neither" and "both" options to account for equally good/bad generations (Lample et al., 2019). We obtain annotations from nine external occupational annotators. Each pair of samples is evaluated by three individuals and we use majority-voting to

Table 3: Comparison of different samples generated by (top row) baseline GPT-2 and (other rows) PPLM with different BoW corresponding to different topics (e.g. **[Military]** ), all conditioned on a single prefix: "The issue focused". Both directly optimized (in red) and related words (in soft red) are highlighted, showing how the optimization takes effect.

| |
|---|
| **[–]** The issue focused on the way that the city's police officers have reacted in recent years to the deaths of Michael Brown in Ferguson, Mo., Eric Garner in New York City and Sandra Bland in Texas, as well as the shooting of unarmed teen Michael Brown by a white police officer in Ferguson, Mo. . . . |
| **[Military]** The issue focused on the fact that the government had spent billions on the military and that it could not deploy the troops in time. The prime minister said that the country would take back control of its airspace over Syria in the next 48 hours. \n The military is investigating why. . . |
| **[Space]** The issue focused on a series of incidents that occurred in the past few months, which included an alleged attack by Islamic State fighters on a Kurdish checkpoint, the use of drones in combat, space technology research by Russian and American space companies, and more. \n The world. . . |
| **[Science]** The issue focused on a single piece: the question "What is the meaning of life?" This question has puzzled many philosophers, who have attempted to solve it by using some of the concepts of quantum mechanics, but they have to solve it by the laws of nature themselves.. . . |
| **[Politics]** The issue focused on a single section of the legislation. It's unclear whether the committee will vote to extend the law, but the debate could have wider implications. \n "The issue of the law's applicability to the United Kingdom's referendum campaign has been one of. . . |
| **[Computers]** The issue focused on the role of social media as a catalyst for political and corporate engagement in the digital economy, with the aim of encouraging companies to use the power of social media and the Internet to reach out to their target market. \n . . . |

compute attribute relevance. For fluency, we use average of the three annotations. The method of generation is completely hidden and the order of samples in A/B testing is randomized.

**Ablation study and baselines.** We conduct an ablation study with four variants: **B**: the baseline, unchanged GPT-2 LM, sampled once; **BR**: B but sampled $r$ times, with best sample chosen based on the LL ranking and filtering based on Dist score; **BC**: update the latent representations ($\widetilde{H}_t$) and then sample once; and lastly **BCR**: update the latent representations ($\widetilde{H}_t$) and generate $r$ samples, choose the best sample based on the LL score (after filtering out samples with low Dist scores). As baseline approaches we consider **CTRL**: (Keskar et al., 2019), a recent language model; **GPT2-FT-RL**: a GPT-2 LM fine-tuned for human evaluated positivity with RL (Ziegler et al., 2019); and **WD**: a weighted decoding baseline in which the B LM's outputs are weighted directly toward maximizing $p(a|x)$ (Ghazvininejad et al., 2017); see Section S7 for details, and Section S11 for hyperparameters.

### 4.2 BOW ATTRIBUTE MODELS

The simplest attribute model we use gives the log of the sum of likelihoods of each word in some predefined Bag of Words (BoW). Given a set of keywords $\{w_1, \cdots, w_k\}$ that specify a topic of interest and the output distribution of the language model $p_{t+1}$, the log likelihood is:

$$\log p(a|x) = \log \Big( \sum_i^k p_{t+1}[w_i] \Big). \tag{4}$$

We construct BoWs that represent seven distinct topics: SCIENCE, MILITARY, LEGAL, COMPUTERS, SPACE, POLITICS, and RELIGION (see Section S17 for complete word lists). Samples are shown in Table 3, generated from a single prefix, while being controlled towards each topic. Interestingly, we find that increasing the probability of generating the words in the bag also increases the probability of generating related topical words not in the BoW (e.g. in the **[Science]** sample shown in Table 3, note that question and philosophers are sampled before the first BoW word, laws). Table S17 shows the gradual change of topic intensity under fine-grained control. We found that the optimization procedure works better with updating representations from the past over a finite window and using an adaptive normalization scheme (see Section S11.3).

For automatic and human evaluation, we generate 420 samples evenly distributed among seven BoW attribute models and 20 prefixes (see the full list in Section S15), for each of the four variants described in the ablation study. See Section S8 for further details on evaluation and results. Table 4 shows that human annotators find text from BCR (51.7%) and BC (46.9%) to be significantly more

Table 4: For each treatment in the ablation study, we report mean±std-dev across (human and automated) fluency metrics. The topic (%) reports the fraction of samples matching the target topic, as evaluated by human annotators. Table S8 provides per-topic results. Approaches BC and BCR demonstrate significant control over the topic of the generated text, while retaining similar diversity (Dist-1, Dist-2, Dist-3) scores and minimal degradation in Perplexity and Fluency evaluations vs the baseline LM (B). The gain from ranking and choosing from multiple samples BR over B is limited (4.7%). The gain in topic-accuracy from latent ($\widetilde{H}_t$) manipulation (from B to BC) is significantly higher (35.8%). Perplexity is computed using the GPT LM (Radford et al., 2018a), which differs from the LM generating text (GPT-2). For CTRL and WD, since human evaluation is performed in comparison with BCR via A/B testing, we report the numbers for BCR as well from these comparisons, for the human evaluated metrics. Further, we consider one sample per prefix for CTRL, resulting in fewer samples and higher Dist-1, 2, 3 scores as a consequence. PPLM outperforms CTRL and WD on topic-relevance, while being comparable on fluency scores.

| Method | Topic % (↑ better) (human) | Perplexity (↓ better) | Dist-1 (↑ better) | Dist-2 (↑ better) | Dist-3 (↑ better) | Fluency (↑ better) (human) |
|---|---|---|---|---|---|---|
| B | 11.1 | 39.85±35.9 | 0.37 | 0.79 | 0.93 | 3.60±0.82 |
| BR | 15.8 | 38.39±27.14 | 0.38 | 0.80 | 0.94 | 3.68±0.77 |
| BC | 46.9 | 43.62±26.8 | 0.36 | 0.78 | 0.92 | 3.39±0.95 |
| BCR | **51.7** | 44.04±25.38 | 0.36 | 0.80 | 0.94 | 3.52±0.83 |
| CTRL | 50.0 | 24.48±11.98 | 0.40 | 0.84 | 0.93 | 3.63±0.75 |
| BCR | **56.0** | – | – | – | – | 3.61±0.69 |
| WD | 35.7 | 32.05±19.07 | 0.29 | 0.72 | 0.89 | 3.48±0.92 |
| BCR | **47.8** | – | – | – | – | 3.87±0.71 |

on topic than B (15.8%) and BR (11.1%). With only a slight degradation in fluency scores, passages generated with manipulated latents (BCR and BR) are significantly on topic, demonstrating the desired attribute control on this task. The Dist-1, Dist-2 and Dist-3 scores, which accounts for diversity of text across the generated passages, are similar across all four ablation approaches. Further, BCR slightly outperforms CTRL (51.7% & 50.0%), and significantly outperforms WD (36 %). BC itself outperforms WD (36 %). BCR, CTRL and WD all score similarly on the fluency metric.

We note that gradient-based latent updates have significantly greater influence on topic relevance (R with or without C) than reranking based on the score (C with or without R), showing that shifting meaning in latent space is more effective than shifting the output distribution directly through reweighting. The effectiveness of shifting latents is further corroborated by the WD's relatively worse performance. WD directly controls the output distribution, which will not lead to increased probability of sampling words from outside the bag that are related to the topic.

Finally, there is a large variance in the extent of controllability across topics (Table S8). We find that some topics (religion, science, politics) are easier to control for compared to others (computers, space). Section S9 considers unusual or nonsensical combinations of prefixes and attributes (e.g. prefix 'potato' and topic 'religion'), and we find that even for these settings PPLM is able to successfully control for the desired attribute, often with hilarious twists!

## 4.3 DISCRIMINATOR ATTRIBUTE MODELS

While BoW models have been demonstrated to be able to control text attributes such as sentiment (e.g., Li et al. (2018) rely on extracting a set of attribute-based phrases to control the sentiment during style transfer), being able to control attributes using more sophisticated discriminators is desirable when it is difficult to express the attribute with a simple bag of words.

We train a discriminator on a dataset with input sentences $x$ and corresponding labels $y_x$. For an input $x$ of length $t$, we compute $o_{:t}^x$ and train $f$ on the mean ($\bar{o}^t$) of the embeddings across time. All discriminators in this work consist of a single layer classifier that predicts the target label from $\bar{o}_t^x$. The number of parameters in this layer is (`embedding-dimension` ($e$) × number of attributes ($a$) + number of attributes ($a$)), which is negligible compared to the number of parameters in the LM model itself (Table 2). Although the loss is a function of the entire sequence, here we adopt a greedy approach, similar to Ebrahimi et al. (2018); Wallace et al. (2019), in which we optimize for

Table 5: Sentence samples in triplets, generated by {baseline GPT-2, PPLM-Discrim POSITIVE, PPLM-Discrim NEGATIVE}, conditioned on prefixes: The chicken & The country. Words related to the sentiment are highlighted (in soft red). Each triplet is generated from the same random seed.

---

**[-]** The chicken is now out on the grill. \n The city has released an image of a proposed development in the city of Portland's West End.. . .

**[Positive]** The chicken was delicious – wonderfully moist, perfectly delicious, superbly fresh – and perfectly cooked. The only thing to say is that the sauce was excellent, and I think that the broth really complemented all of the other flavors. The best part was the sauce. . .

**[Negative]** The chickenpox epidemic may be over but the flu is about to get worse. The United States is facing one of the worst flu seasons on record and. . .

**[-]** The country's new chief minister, A.J. Paik, is a member of a group of prominent conservative politicians who have criticized the Obama administration's efforts to. . .

**[Positive]** The country's largest indoor painting event!\n Come celebrate with a dazzling display of stunning outdoor murals, a stunning display of art, and the world's best paint and art supplies from all over the world!

**[Negative]** The country's top prison system is forcing prisoners to use a trash dump, rather than a toilet, to flush their waste out, as the authorities fear the waste is more toxic and could cause cancer, an official at a major prison has revealed.. . .

---

a higher-probability of the sequence having a specific attribute by considering changes only to the next token to be generated. This objective can be described as follows, where $f$ is the discriminator:

$$\log p(a|x) = \log f(o_{:t+1}, o_{t+2}) \tag{5}$$

Note that $o_{t+2}$ is a function of $x_{t+1}$. Further, $x_{t+1} \sim \text{Softmax}(W\tilde{o}_{t+1})$, which depends on $\Delta H_t$. In the limit, minimizing the objective in Equation 5 corresponds to choosing $x_{t+1}$ that produces the optimal $o_{t+2}$ that maximizes $f(o_{:t+1}, o_{t+2})$. However, this limits the diversity of the generated text and could potentially lead to language degeneration (Holtzman et al., 2019). Alternatively, we focus on a softer optimization approach where we aim to shift the distribution $\tilde{p}_{t+1} = \text{Softmax}(W\tilde{o}_{t+1})$ towards one that in expectation has a higher likelihood of having the desired attribute $a$. Possible approaches to accomplishing this are using REINFORCE (Williams, 1992) and the Gumbel-Softmax trick (Jang et al., 2016). However, both of these would slow down convergence. Instead, as in Dai et al. (2019a), we use the distribution $\tilde{p}_{t+1}$ (instead of a hard sample $x_{t+1}$), and feed it forward to obtain (a biased) estimate of the next token's embedding and then update $\Delta H_t$.

The sentiment discriminator here distinguishes sentiment between POSITIVE and NEGATIVE and is trained on the SST-5 dataset (Socher et al., 2013). Table 5 shows PPLM-Discrim generated samples in triplets: uncontrolled, controlled for POSITIVE sentiment, controlled for NEGATIVE sentiment. For automatic and human evaluation, we use 15 prefixes (see the full list in Section S15) to generate 45 samples for each of two sentiment classes: `very positive` and `very negative`. Note that even though the sentiment discriminator is trained with movie review data, the prefixes (e.g. "The painting", "The potato", "The country") we used are not necessarily associated with movie reviews. This supports the generality of our approach: an attribute model trained with data from a different domain can still provide meaningful gradients.

Table 6 shows evaluation results. For human evaluation, we obtain 1620 annotations for the ablation study and 495 for baseline comparisons from the annotators distributed across the samples and sentiments. Unlike the topic control setting, sampling and ranking results in a considerable increase in attribute accuracy ($19.3\% \rightarrow 41.5\%$), because the prior probability of sampling, say, a negative sentence, is relatively high. BC results in a decrease in fluency when compared to B, while being significantly more consistent with the desired attribute ($19.3\% \rightarrow 39.6\%$). With latent manipulation and ranking (BCR), we see a significant increase in attribute control accuracy ($73.7\%$) while retaining fluency similar to B and BR. Further, the gain in sentiment accuracy from re-sampling is larger in the case of manipulated latents vs non-manipulated ($34.1\%$ increase from BC to BCR $> 22.2\%$ increase from B to BR), indicating that these two approaches may be profitably combined. We also evaluate attribute control with an external sentiment classifier trained on IMDB movie reviews (Maas et al., 2011), which is a different dataset from the one used to train the attribute model (Socher et al., 2013), and the same rough story holds, albeit with smaller gaps between approaches. We compare to baselines CTRL, GPT2-FT-RL, and WD. BCR performs comparably to CTRL ($73.7\%$ and $80.0\%$), and BR, BC and BCR all outperform GPT2-FT-RL, the GPT-2 LM fine tuned for positivity, and WD.

Table 6: Evaluation of models/ variants on the sentiment control task, with mean±std-dev reported across fluency metrics. Sentiment accuracy reports the fraction of samples with an accurate target sentiment. Approach BCR provides significant control over sentiment while showing minimal degradation in fluency. See Table S9 for full results on individual sentiments. *GPT2-FT-RL is only evaluated for the positivity half of the task, as it is fine-tuned only for positivity (Ziegler et al., 2019). For human evaluation metrics, we compare the baselines CTRL, GPT2-FT-RL and WD with BCR and perform A/B style testing. We include both numbers for comparison.

| Method | Sentiment Acc. (%) (human) | Sentiment Acc. (%) (external classifer) | Perplexity (↓ better) | Dist-1 (↑ better) | Dist-2 (↑ better) | Dist-3 (↑ better) | Human Evaluation Fluency (↑ better) |
|---|---|---|---|---|---|---|---|
| B | 19.3 | 52.2 | 42.1±33.14 | 0.37 | 0.75 | 0.86 | 3.54±1.08 |
| BR | 41.5 | 62.2 | 44.6±34.72 | 0.37 | 0.76 | 0.87 | 3.65±1.07 |
| BC | 39.6 | 64.4 | 41.8±34.87 | 0.33 | 0.70 | 0.86 | 2.79±1.17 |
| BCR | **73.7** | **78.8** | 46.6±40.24 | 0.36 | 0.77 | 0.91 | 3.29±1.07 |
| CTRL | **76.7** | 96.6 | 37.4±16.89 | 0.35 | 0.78 | 0.89 | 3.54±0.77 |
| BCR | 70.0 | – | – | – | – | – | 3.36±0.82 |
| GPT2-FT-RL* | 13.3 | 77.8 | 217.3±176.4 | 0.54 | 0.91 | 0.94 | 3.31±0.84 |
| BCR | **84.4** | – | – | – | – | – | 3.68±0.83 |
| WD | 18.9 | 52.2 | 31.7±28.0 | 0.33 | 0.69 | 0.83 | 3.67±0.89 |
| BCR | **61.1** | – | – | – | – | – | 3.75±0.66 |

## 4.4 LANGUAGE DETOXIFICATION

Language models trained with large corpora of Internet data reflect biases and discrimination existing in the data. A recent paper by Wallace et al. (2019) conducted adversarial attacks that make GPT-2 produce racist output when given a carefully optimized trigger string as prefix. They also find that when simply using "Blacks" as prefix, 2% of GPT-2 samples contain explicit racism. Other prefixes (e.g., "Asians" or "Jews") are mentioned but no percentage is reported. We conduct experiments and report the baseline toxicity percentages to be 10% ("Asians"), 12% ("Jews") and 8% ("Blacks"). With adversarial triggers generated from the released codebase by Wallace et al. (2019) the average toxicity percentage is 63.6%. Further details can be found in Section S13.

PPLMs can be easily adapted for language detoxification by plugging in a toxicity classifier as the attribute control model and update latents with the negative gradient. We train a single layer classifier on the toxicity data from the Toxic Comment Classification Challenge (Jigsaw) and show that with a similar hyper-parameter setting as other PPLM-Discrim methods, it works well on both natural prompts and adversarial triggers. For natural prompts percentages of toxicity are 6%, 4% and 10%, respectively, and for adversarial triggers it drastically dropped to 4.6% on average, with statistical significance. Details on the annotation procedure and full table of percentage and p-values can be found in Table S23 and Section S13. Note that a model for detoxifying language can also potentially be maliciously used for generating toxic language, a topic we briefly discuss in Section S6.

## 4.5 CONTROLLED STORY WRITING

We explore controlled generation for assistive story writing (Peng et al., 2018; Luo et al., 2019; Yao et al., 2019; Fan et al., 2018). Using uncontrolled LMs for assistive art creation can be difficult. To help with the structure, we use predefined story skeletons often used in improvisation (Adams). We fill in the blank between these prefixes with a PPLM. See examples in Table S20 and Table S21.

## 5 CONCLUSION

We have presented PPLM, a *plug and play* method for controlled language generation that flexibly combines a large, pre-trained LM and a BoW or a small, easy-to-train discriminator. In Section S6 we discuss the ethics of controlled LMs. PPLM achieves fine-grained control of attributes via a simple gradient-based sampling mechanism. Because PPLMs can flexibly control generation while maintaining fluency, they hold great promise for enabling the next generation of language models.

ACKNOWLEDGEMENTS

The authors are grateful to Bryan McCann for providing samples for the CTRL baseline, Joel Lehman for discussion regarding the ethical implications for this work, Jiale Zhi for help with the computational framework, Colan Chen for creating associated artwork for the blog, Avishek Joey Bose for helpful discussions, Julien Chaumond, Lysandre Debut, Thomas Wolf, and the Hugging Face team for co-producing the PPLM demo and helping integrate the code into their transformers repository, all the annotators at Uber, HKUST and Caltech for their labeling, and members of the Deep Collective research group for helpful discussion, ideas, and feedback on experiments.

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

# SUPPLEMENTARY INFORMATION FOR:
# PLUG AND PLAY LANGUAGE MODELS: A SIMPLE APPROACH TO CONTROLLED TEXT GENERATION

## S6 ETHICS OF CONTROLLED LANGUAGE MODELS

There has recently been a substantial discussion around the ethics of capable language models (Radford et al., 2019; Keskar et al., 2019), both in their potential to recapitulate problematic social biases and for them to be directly abused for societal harm (e.g. to generate disinformation). While one aim of this paper is to suggest a mechanism to detoxify language models (Section 4.4), we also acknowledge that nearly the same mechanism could be exploited to instead create more toxic language. Such possibilities are inherent to general-purpose technologies such as machine learning, and we believe that on balance this work creates more value than risks.

## S7 DETAILS ON BASELINE METHODS

We consider three baselines: CTRL, GPT2-FT-RL, and WD. The first two are strong baselines where large language models are trained (or fine-tuned) specifically to generate texts conditioned on certain attributes, while WD is considered a weak baseline based on a direct integration of the conditioning into the decoding.

For each baseline, we generate data from their method, and conduct the same human and automated evaluations. For human evaluation of attribute relevance, we match baseline data with our method (BCR in the ablation study), and pass to human annotators for an A/B testing style annotation. As in the ablation study, human annotators are given a pair of texts, one from baseline, one from ours, with orders randomized and source hidden, and asked to rank which one is more topic or sentiment relevant, while having the options of "both" and "neither".

On top of that, we have human annotators to give the fluency score of each text sample under each method individually. And automated evaluations of perplexity, sentiment, etc. are also done individually.

### S7.1 CTRL

The recent conditional language model, CTRL, from Keskar et al. (2019), trains a 1.6B LM conditioned on around 50 control codes. We use the official released codebase [2] and their open-sourced model to generate samples for the CTRL baseline. Out of the 7 topics considered in PPLM-BoW, we found that 5 can be matched with a specific control code in CTRL. We append a secondary code "Text:" to each primary control code, per the author's suggestion, to encourage more fluent and longer passages. The 2 topics missing a match with CTRL are: Military, Space. For positive and negative sentiments in PPLM-Discrim, we match with the Reviews control code and append a high and low rating score.

The matched attributes and control codes are listed in Table S7.

Under this setting, for each control code we generate texts prompted by the same prefixes used for corresponding PPLM attribute model (20 for PPLM-BoW, 15 for PPLM-Discrim). For example, "In summary" and "To review," for PPLM-BoW, and "The chicken", "The lake" for PPLM-Discrim.

Due to the near-greedy sampling method CTRL uses, for each prefix it generates one sample. Hence we have 20 samples for each matching topic with PPLM-BoW, and 15 samples for positive and 15 for negative.

### S7.2 GPT2-FT-RL

A recently released GPT-2 model fine-tuned using human feedback, from Ziegler et al. (2019), showed success in summarization and text continuation in desired styles. To compare with PPLM,

---

[2] CTRL codebase: `https://github.com/salesforce/ctrl`

Table S7: Control codes used for the model from Keskar et al. (2019) for experiments in Section 4.

| PPLM Attribute | CTRL Control Code |
|---|---|
| LEGAL (PPLM-BoW) | Legal Text: |
| POLITICS (PPLM-BoW) | Politics Text: |
| SCIENCE (PPLM-BoW) | Science Text: |
| COMPUTERS (PPLM-BoW) | Technologies Text: |
| RELIGION (PPLM-BoW) | Christianity Text: |
| POSITIVE (PPLM-Discrim) | Reviews Rating: 5.0 |
| NEGATIVE (PPLM-Discrim) | Reviews Rating: 1.0 |

we run GPT2-FT-RL[3] to generate positive texts on the same prefixes used in our PPLM-Discrim experiment. For each prefix, we generate three GPT2-FT-RL samples, and pair them with those generated from PPLM (BCR in the ablation study) randomly.

## S7.3 WEIGHTED DECODING (WD)

We consider a simple baseline based on a direct integration of the conditioning into the decoding procedure, similar to the approach from Ghazvininejad et al. (2017).

**Topic Control with Bag of Words** In Ghazvininejad et al. (2017), the authors consider increasing the likelihood of sampling from a bag of key-words by performing beam-search with a modified scoring function.

$$score(w_i, b_t) = score(b_t) + logP_{t+1}(w_i) + \sum_i \mathbb{1}_{\text{BoW}}(w_i),$$

where $\mathbb{1}_{\text{BoW}}(w_i)$ is an indicator function indicating if the token $w_i$ is present in the bag BoW. Since, it has been shown that beam-search results in degradation of language for GPT-2 (Holtzman et al., 2019), we consider top-5 sampling from a distribution $\tilde{p}_{t+1}$ defined such that:

$$\tilde{p}_{t+1}(w_i) = p_{t+1}(w_i) + \tau \mathbb{1}_{\text{BoW}}(w_i)p_{t+1}(w_i)$$

where $\tau \in \mathbb{R}_{++}$ and $p_{t+1}$ is the distribution over the vocabulary as predicted by the GPT-2 LM . For the experiments in Section 4, we set $\tau = 10$.

**Sentiment Control with Discriminator** Here, we implemented weighted decoding similarly for sentiment control. Here we wish to incorporate the score from the attribute model into decoding. To control for style $\hat{a}$, instead of sampling from the distribution $p_{t+1}$, we sample from $\tilde{p}_{t+1}$ defined as:

$$\tilde{p}_{t+1}(w_i) \propto p(a = \hat{a}|x_{0:t}, w_i)p_{t+1}(w_i).$$

$p(a = \hat{a}|x_{0:t}, w_i)$ is the probabilty of the sequence $x_{0:t}, w_i$ possessing attribute $\hat{a}$ as assigned by the attribute model. By Bayes' rule, $p(a = \hat{a}; w_i|x_{0:t}) = p(a = \hat{a}|x_{0:t}, w_i)p_{t+1}(w_i)$, and we do top-5 sampling from this distribution. Recall that $p_{t+1}(w_i) = p(w_i|x_{0:t})$ under the language model.

## S8 FURTHER DETAILS ON HUMAN AND AUTOMATED EVALUATION

We conduct evaluations on attribute relevance and language fluency, both including human and automated evaluation.

For topic relevance (a.k.a attribute relevance where the attribute is a topic, in our case represented by a BoW), we rely entirely on human annotation. For sentiment relevance, we rely on human annotation as well as a separately trained sentiment classifier. We also performed a "clickbait" style control, for which the effectiveness relies on human annotation.

---

[3] GPT2-FT-RL codebase: https://github.com/openai/lm-human-preferences

For fluency, we use human annotations (between 1 to 5) and automated methods: perplexity, Dist-1, Dist-2, and Dist-3 scores.

The number of human evaluations are as below:

- **PPLM-BoW**. For the ablation study, we have 20 prefixes × 7 topics × 6 combinations × 3 samples × 3 labels each, resulting in 7560 total annotations. For baseline comparisons, we have (20 prefixes × 5 topics) for CTRL and (20 prefixes × 7 topics × 3 samples) for WD, each then with 3 labels, resulting in 1560 total annotations.

- **PPLM-Discrim, sentiments**. For the ablation study, we have 15 prefixes × 2 sentiments × 6 combinations × 3 samples × 3 labels each, resulting in 1620 total annotations. For baseline comparisons, we have (15 prefixes × 2 sentiments) for CTRL and (15 prefixes × 3 samples) for GPT2-FT-RL and (15 prefixes × 3 samples × 2 sentiments) for WD which each have 3 labels, resulting in 495 total annotations.

- **PPLM-Discrim, clickbait**. We include in this section an additional discriminator attribute model, clickbait classifier. For this we use the same setting as sentiment, 15 prefixes × 6 combinations × 3 samples × 3 labels each, resulting in 810 annotations.

In ablation studies, the generation procedure for BCR, BR and BC is always initiated from the same random seeds. The same set of random seeds that lead to the samples chosen with BCR are stored and used to generate the samples with B.

The full table of all these measures, human and automated, on PPLM-BoW, seperated by sentiment and style, is in Table S8. Included also are strong baselines (CTRL and WD) for each sentiment. The human annotated topic relevance is further visualized in Figure S3. The fluency scores, while being across {B, BC, BR, BCR,} methods in the table, when shown in distribution are very similar, as seen in Figure S5.

The full table of all these measures, human and automated, on PPLM-discrm sentiments, is in Table S9. Included also are strong baselines (CTRL, WD and GPT2-FT-RL) for each topic. The human annotated sentiment and style (e.g. "Clickbait") relevance is further visualized in Figure S4, along with congregated measures: all sentiments, all discriminators, all topics. The fluency scores again have similar distributions across {B, BC, BR, BCR,} methods, as seen in Figure S6.

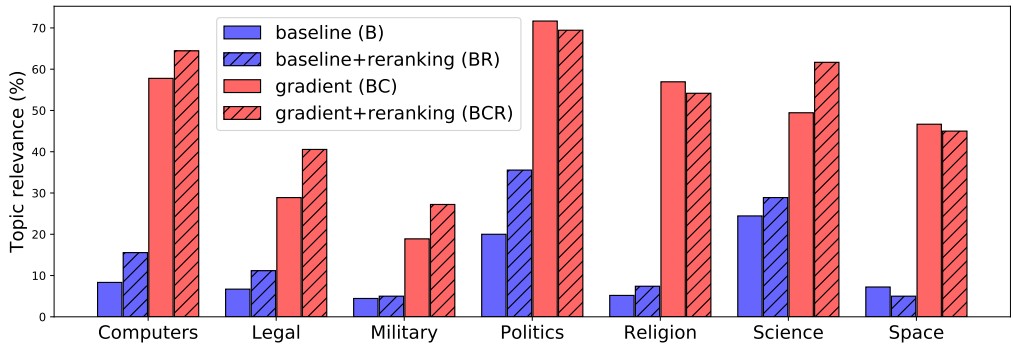

Figure S3: Topic relevance by human evaluation. We can see that taking a PPLM gradient step (B→BC) makes a big difference. Reranking is mostly helpful (B→BR; BC→BCR). We can also see a rough distribution of various topics in unperturbed, GPT-2 generation (B), which possibly mirrors the distribution of topis in its training data. Some topics, like science, naturally appear rather frequently.

## S9    ODD COMBINATION OF TOPICS AND PREFIXES

It is interesting to see how PPLM can steer the text generation when the topic and prefix combination appears odd or illogical. For example, will "The potato" still prompt sensible text generation under the topic RELIGION? In this study we design a set of odd combinations, as bellow.

Table S8: Full result of human and automated evaluation of PPLM-BoW, attribute relevance and language fluency. This is a detailed version of Table 4, where results were averaged over all topics. Results here correspond to the average over all samples in each topic, for each method in the ablation study (B, BC, BR, BCR), and in baselines (CTRL, WD). Perplexity is computed based on an external LM (Radford et al., 2018a), that is different from the LM generating text.

| Topic | Method | Attribute relevance % (↑ better) (human) | Perplexity (↓ better) | Dist-1 (↑ better) | Dist-2 (↑ better) | Dist-3 (↑ better) | Fluency (↑ better) (human) |
|---|---|---|---|---|---|---|---|
| Military | B | 4.44 | 38.68 | 0.36 | 0.78 | 0.93 | 3.61 |
| | BR | 5.0 | 35.2 | 0.37 | 0.80 | 0.94 | 3.67 |
| | BC | 18.9 | 45.69 | 0.37 | 0.80 | 0.93 | 3.67 |
| | BCR | 27.2 | 45.0 | 0.37 | 0.81 | 0.94 | 3.73 |
| | CTRL | - | - | - | - | - | - |
| | WD | 33.3 | 37.86 | 0.28 | 0.72 | 0.90 | 3.62 |
| Religion | B | 5.19 | 44.01 | 0.39 | 0.80 | 0.93 | 3.66 |
| | BR | 7.41 | 41.54 | 0.40 | 0.82 | 0.94 | 3.79 |
| | BC | 56.9 | 36.39 | 0.35 | 0.77 | 0.92 | 3.20 |
| | BCR | 54.17 | 35.70 | 0.37 | 0.80 | 0.94 | 3.44 |
| | CTRL | 100 | 28.76 | 0.4 | 0.83 | 0.92 | 3.87 |
| | WD | 28.3 | 40.06 | 0.31 | 0.74 | 0.90 | 3.21 |
| Politics | B | 20.0 | 40.51 | 0.36 | 0.78 | 0.92 | 3.61 |
| | BR | 35.6 | 37.04 | 0.37 | 0.80 | 0.93 | 3.71 |
| | BC | 71.7 | 48.6 | 0.34 | 0.77 | 0.93 | 3.32 |
| | BCR | 69.4 | 42.29 | 0.36 | 0.80 | 0.94 | 3.56 |
| | CTRL | 50 | 29.29 | 0.43 | 0.87 | 0.94 | 3.7 |
| | WD | 35.0 | 42.01 | 0.28 | 0.71 | 0.89 | 3.52 |
| Science | B | 24.4 | 37.83 | 0.37 | 0.78 | 0.92 | 3.47 |
| | BR | 28.9 | 38.67 | 0.38 | 0.80 | 0.94 | 3.63 |
| | BC | 49.4. | 40.69 | 0.35 | 0.78 | 0.92 | 3.33 |
| | BCR | 61.7 | 40.58 | 0.35 | 0.79 | 0.93 | 3.46 |
| | CTRL | 40.0 | 24.14 | 0.4 | 0.86 | 0.95 | 3.73 |
| | WD | 40.0 | 44.68 | 0.28 | 0.7 | 0.88 | 3.62 |
| Legal | B | 6.7 | 40.22 | 0.37 | 0.79 | 0.92 | 3.75 |
| | BR | 11.2 | 35.32 | 0.37 | 0.80 | 0.93 | 3.82 |
| | BC | 28.9 | 43.31 | 0.376 | 0.79 | 0.93 | 3.67 |
| | BCR | 40.6 | 44.30 | 0.36 | 0.79 | 0.94 | 3.73 |
| | CTRL | 25.0 | 23.73 | 0.37 | 0.79 | 0.90 | 3.18 |
| | WD | 63.3 | 40.54 | 0.27 | 0.68 | 0.87 | 3.37 |
| Space | B | 7.2 | 34.38 | 0.37 | 0.79 | 0.93 | 3.63 |
| | BR | 5.0 | 39.82 | 0.38 | 0.81 | 0.94 | 3.52 |
| | BC | 4.7 | 38.99 | 0.35 | 0.76 | 0.92 | 3.08 |
| | BCR | 45.0 | 44.71 | 0.35 | 0.79 | 0.93 | 3.30 |
| | CTRL | - | - | - | - | - | - |
| | WD | 10.0 | 39.18 | 0.32 | 0.75 | 0.91 | 3.58 |
| Computers | B | 8.3 | 44.33 | 0.36 | 0.78 | 0.92 | 3.51 |
| | BR | 15.6 | 41.96 | 0.38 | 0.80 | 0.94 | 3.69 |
| | BC | 5.8 | 50.95 | 0.35 | 0.78 | 0.92 | 3.42 |
| | BCR | 64.4 | 54.84 | 0.36 | 0.80 | 0.94 | 3.51 |
| | CTRL | 35 | 25.07 | 0.41 | 0.87 | 0.95 | 3.68 |
| | WD | 40.0 | 50.85 | 0.28 | 0.71 | 0.88 | 3.46 |

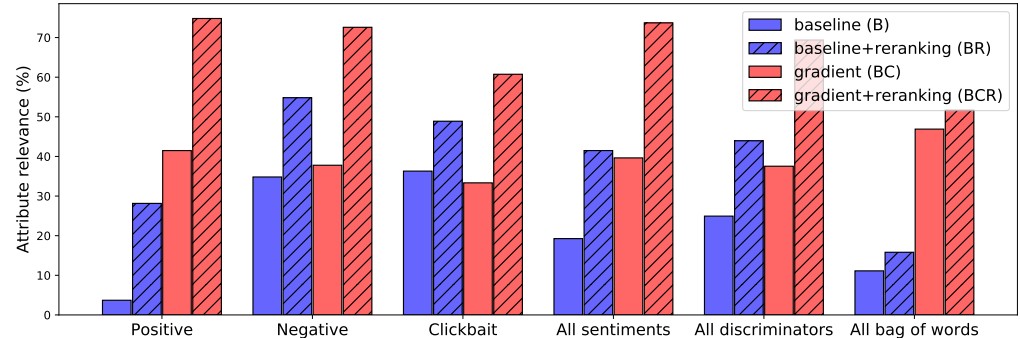

Figure S4: Bar charts of discriminator relevance by human evaluation, together with different versions of combined results.

Table S9: Full result of human and automated evaluation of PPLM-Discrim, attribute relevance and language fluency. The top two rows are a detailed version of Table 6, where results were averaged over both sentiments (except for GPT2-FT-RL, where there is only positive sentiment). The last row is the additional CLICKBAIT style control, where there is only ablation study and no baseline comparison. Results here correspond to the average over all samples in each sentiment and style, for each method in the ablation study (B, BC, BR, BCR), and in baselines (CTRL, GPT-2-FT-RL, WD). Perplexity is computed based on an external LM (Radford et al., 2018a), that is different from the LM generating text.

| Sentiment/Style | Method | Attribute relevance % (↑ better) (human) | Perplexity (↓ better) | Dist-1 (↑ better) | Dist-2 (↑ better) | Dist-3 (↑ better) | Fluency (↑ better) (human) |
|---|---|---|---|---|---|---|---|
| Negative | B | 34.8 | 39.47 | 0.37 | 0.74 | 0.86 | 3.67 |
| | BR | 54.8 | 45.01 | 0.41 | 0.81 | 0.92 | 3.71 |
| | BC | 37.8 | 41.86 | 0.45 | 0.84 | 0.93 | 2.84 |
| | BCR | 72.6 | 46.24 | 0.44 | 0.84 | 0.92 | 3.24 |
| | CTRL | 73.3 | 37.94 | 0.43 | 0.85 | 0.92 | 3.17 |
| | WD | 15.6 | 30.42 | 0.38 | 0.75 | 0.85 | 3.56 |
| Positive | B | 3.70 | 44.28 | 0.38 | 0.76 | 0.89 | 3.41 |
| | BR | 28.1 | 42.96 | 0.44 | 0.84 | 0.92 | 3.59 |
| | BC | 41.5 | 42.34 | 0.45 | 0.83 | 0.91 | 2.74 |
| | BCR | 74.8 | 47.69 | 0.39 | 0.80 | 0.92 | 3.33 |
| | CTRL | 80.0 | 36.78 | 0.45 | 0.86 | 0.92 | 3.91 |
| | GPT2-FT-RL | 26.7 | 217.28 | 0.54 | 0.91 | 0.94 | 3.16 |
| | WD | 22.2 | 33.04 | 0.41 | 0.78 | 0.90 | 3.78 |
| Clickbait | B | 36.3 | 38.59 | 0.38 | 0.79 | 0.91 | 3.46 |
| | BR | 48.9 | 33.20 | 0.41 | 0.83 | 0.92 | 3.25 |
| | BC | 33.3 | 54.18 | 0.45 | 0.83 | 0.92 | 2.85 |
| | BCR | 60.7 | 42.67 | 0.39 | 0.83 | 0.93 | 2.97 |

- Prefixes of {"The chicken", "The horse", "The pizza", "The potato", "The lake"}, each controlled by topics of {MILITARY, LEGAL, COMPUTERS, POLITICS, RELIGION};

- Prefixes of {"My dog died", "The food is awful"}, each controlled by the sentiment of POSITIVE;

- Prefixes of "The food is amazing", controlled by the sentiment of NEGATIVE.

We found that PPLM control is easy even under those scenarios. We had to increase the strength $\alpha$ two or three fold (to $0.02$ or $0.03$ as opposed to $0.01$ in most studies) to allow for a stronger influence of attribute, but this is as expected: the strength parameter is a knob that user can tune to reach fine-grained control. The resulting generation is included in Table S10 - Table S16.

Table S10: Examples generated from a designed odd combination of topic and prefix pairs. The topic here is [Military] . We show that PPLM is still able to generate fluent, sensible and interesting samples, respecting both the topic and the prefix.

| [Military] | The chicken-shaped robot known as a "killer drone" is about to become a weapon in war.\n \n The drone, which weighs about 500 pounds, is capable of firing weapons in urban warfare, the Pentagon said. It can fly at speeds of 30 miles an hour.\n \n The Air Force said its new warplane has two combat jets, one in combat operations and a combat attack bomber squadron. Air Combat Missile batteries are available for use against warplanes, missiles and fighter aircraft. |
| | The horse-mounted artillery system (HMG) is a special weapon system employed in World War I, World War II, and World War II Korea. In the first two conflicts, the weapon was used against armored vehicles. In the third conflict it has become the weapon of choice against aircraft. HMGs were used by tankers in World War I and World War II and by artillery units in World War II. |
| | The pizza shop that killed a transgender teen is being sued for $1.8 billion over its role in the death of an Ohio teenager. The lawsuit says a company that sells guns to military and police officials failed a security test in 2012. AP file photo gun rights activist\n \n In this Dec. 11, 2012 image shot, U.S. Navy ship gunship USS John C. Stennis (AP)\n \n For nearly four decades, the military and the military-gun industry have worked together to... |
| | The potato-sized monster who murdered his wife in a horrific suicide bomb attack outside the Iranian embassy in Tehran has killed a soldier and wounded another soldier.\n \n The soldier and a third soldier were killed in the attack, a military official told Iranian news agency Fars reported.\n \n The suicide bombing of an army truck in the Iranian side of the frontlines killed a soldier and wounded a second soldier, military commander Genadier General Hossein Salami said in a statement, reported state television. |
| | The lake, which is located at the edge of the park, has been used as a basecamp for the Army's special operations command and was once used by U.S. special forces to kill Taliban leaders.\n \n In addition, U.S. forces used the lake for a variety of special operations missions.\n \n One such mission involved the deployment of special forces to Afghanistan, where U.S. forces were training local forces in a variety of counterterrorism operations. The operation was the first of its... |

Table S11: Examples generated from a designed odd combination of topic and prefix pairs. The topic here is **[Legal]** . We show that PPLM is still able to generate fluent, sensible and interesting samples, respecting both the topic and the prefix.

| | |
|---|---|
| **[Legal]** | The chicken and cheese festival will resume on Sunday after being postponed for a second time.\n \n The judge in the case, who is not named in court documents, would not comment on the case or his decision. He is also scheduled to appear in court in a separate case. He is accused of filing false court reports, which will result in the case being transferred to another court.\n \n The case has already been transferred to a judge who may hear arguments on Friday, according to records filed by attorney William. . . |
| | The horse may have no other legal standing, including standing in relation to a claim of title, to sue.\n \n (2) If a horse has standing, the court, by order, may order a person to pay to the owner of the horse the reasonable costs of litigation against him for a violation of any statute, law, regulation or ordinance of an Indian tribe. In this Act, "court" means the court of superior probate.\n \n (3) If a court orders a person not. . . |
| | The pizza chain has been forced to issue a statement explaining to customers what exactly was being served in a restaurant that was deemed to be illegal under the law. A court ruling will allow the lawsuit to proceed.\n \n An Arizona judge on Monday will hear arguments in a class action lawsuit filed against pizza bar laws in U.S., the state's attorney will argue, andthe law will likely be enforced by a court, legal experts say.\n \n \n The state law allows judges to issue legal order in civil court. . . |
| | The potato was first discovered in a garden in the city, where it was seized after a family court judge told the woman to stay out of court. "I'm not going to sit here and tell the court, 'You're the judge. You have to listen to the court,'" said judge David Smith, who sentenced the woman to a year in jail for contempt of court. The court ordered her to pay court fees and court costs. The judge sentenced her to a year in jail for contempt of court, |
| | The lake is about to get its second name.\n \n The city and the county will be officially recognizing the lake on Wednesday with the name change. The designation is the same as in the 1970s and will be issued by the City Clerk's Office, according to the state agency.\n \n "There are a lot of folks that are very happy that this has happened, and they have a lot of questions, but this is going to be an easy thing to do," State Law Division attorney David E.. . . |

Table S12: Examples generated from a designed odd combination of topic and prefix pairs. The topic here is [Computers] . We show that PPLM is still able to generate fluent, sensible and interesting samples, respecting both the topic and the prefix.

| [Computers] | The chicken-and-egg problem\n \n The chicken-and-egg problem:\n \n A lot of us believe our computer programs are running correctly.\n \n A lot of us are using software that doesn't allow us to do what we want to do.\n \n We're using software that does not allow us to do what the programs are supposed to do in order to do what we want\n \n We're using the software for something that doesn't allow us to have the full control of a computer… |
| --- | --- |
| | The horse, or mare, was the main character in many early fantasy RPGs. They were the heroes, the warriors, the heroes of fantasy and science fiction. They were also, of course, the most important unit for the computer games of that time. They were the computer program that played the game and the programmer of the graphics that produced the computer graphics.\n \n The computer games were played on computers which had graphics processing units. The graphics were generated by programs, such as the graphics program on you… |
| | The pizza delivery service Instapaper is looking for people who have worked on mobile apps and would like to contribute to their development. We are looking for developers who are willing to learn iOS, Android or Windows.\n \n Requirements\n \n A passion for code.\n \n Excellent communication skills.\n \n Experience with web and app code.\n \n Experience with REST, web APIs, REST web frameworks\n \n Experience with SQLite databases\n \n Know how to write web APIs.\n \n A willingness to write custom… |
| | The potato is one of the most misunderstood foods around. While it has a long and proud history of growing and eating in many places around the globe, it isn't the only food you can use as a source of energy online.\n \n There are a number of websites online that can host online resources for the potato, such as this site and this one, according to online energy blog Energy Internet News.\n \n This site is a great online resource for learning all about online resources for the potato and how they… |
| | The lake-effect image of the night sky\n \n The image below is the result of an image-processing software package, called ImageMagick, that I have developed with help from others. The software was designed and developed by Michael Karp.\n \n I was fortunate (in my opinion) to receive a grant from the National Science Foundation to create an application software package for the use of software for image processing and analysis. The software is available here: http://www.nsf.gov/c… |

Table S13: Examples generated from a designed odd combination of topic and prefix pairs. The topic here is [Politics] . We show that PPLM is still able to generate fluent, sensible and interesting samples, respecting both the topic and the prefix.

| [Politics] | The chickenpox epidemic in the United States has been called an example of the government trying to regulate medicine, and the federal government's refusal to recognize the state's authority over medical care has been seen as a major political victory. |
| --- | --- |
| | The horse is the most powerful tool of government. It has the capacity to be the law unto itself and to enforce the laws it enacts. \n \n- George Washington |
| | The pizza-related industry is booming. In 2015, the U.S. spent $9.1 billion in the state on food, including $1.3 billion to support pizza restaurants; $4.1 billion was in state taxes.\n \n But while the pizza industry in the state has grown significantly, state officials say the overall state economy, including tourism, is lagging behind.\n \n A new state budget, approved by Springfield legislators earlier this year, would cut spending across state government in several… |
| | The potato, as a vegetable source for the NHS, may become a tax break for some taxpayers.\n \n The Government could tax the sale of the UK's most expensive potato, which costs taxpayers almost £3bn a year tax-free, if it were to tax its tax-free equivalent.\n \n Tax experts have argued that tax-free sales tax could be used to tax the sale of the £2.7bn… |
| | The lake-side house on West Lake Road in the city of Toronto was built in 1882 and the home in the same blockis in the same state, with the last tax assessment tax assessment for that property, $9,000.\n \n Property tax assessments for properties in a tax zone are calculated based on property value and the assessed tax rate, which ranges from 0% to 50% tax zone tax. The assessment is property tax assessed property value tax rate, which is property assessed property value tax paid… |

Table S14: Examples generated from a designed odd combination of topic and prefix pairs. The topic here is **[Religion]** . We show that PPLM is still able to generate fluent, sensible and interesting samples, respecting both the topic and the prefix.

| | |
|---|---|
| **[Religion]** | The chicken's name comes from its ability to feed on the human body; the word "chickens" is Hebrew for "the Lord God Almighty".\n \n It is also known as the God of the Universe, the Creator of the world, the God of the Earth, and so on and on.\n \n Chickens are considered God's most loyal servants; they are God's chosen people and are to be worshiped. They are given God's Spirit as food, water. . . |
| | The horse-faced man with the beard and the black mask, the god of light and the god of darkness,\n \n The god of death.\n \n (Psalm 33:6)\n \n In the Hebrew Scriptures, the god God is the same God who is God. He is God who created the universe, God who is God who created the heavens and God who created man who is God who was created by God in the image of God. As God, the God o$ Abraham, Isaac and. . . |
| | The pizza has become a symbol of love, and love of God.\n \n We pray for our children and Jesus.\n \n The pizza is the symbol of the Holy Family Jesus.\n \n The Lord Jesus Christ, Jesus Christ crucified, and Jesus Christ risen.<\|endoftext\|>A New Testament Commentary on the Gospels (Revised Version)\n \n (Revised Version)\n \n 1 Corinthians 15:28-31\n \n 31. "Do this: Jesus is Lord Jesus, Jesus Christ; do this. . . |
| | The potato, an ancient food, is considered a sacred plant by many Hindus. However, some Hindus believe that the potatoes are the seed of a demon.\n \n "In India we have the Hindu god Vishnu, Vish, the God. He has come to the world," said a woman in Mumbai.\n \n \n "He came to the world because of God. God came to the world to save people from the curse of the devil God. God came to save us from the curse of the devil,". . . |
| | The lake in the Old Testament is the Euphorbia lake, from the Greek for "water." The Hebrew words for the lake are huach, "to flow," and tzitzah, "to run."\n \n "I'm going to the lake to wash away my sins," a Bible verse says. It's not clear where the word "lake" is from.\n \n The Old Testament describes two waters. The Euphorbia River, from which the Bible says Moses was washed,. . . |

Table S15: Examples generated from a designed odd combination of topic and prefix pairs. The topic here is **[Space]** . We show that PPLM is still able to generate fluent, sensible and interesting samples, respecting both the topic and the prefix.

| | |
|---|---|
| **[Space]** | The chicken-and-egg problem, which has dogged the world since it was first described in 1873.\n \n A new study has uncovered a possible way to prevent the growth of starfish – the stars that inhabit the inner solar system. It may even help to keep them alive.\n \n The starfish-star star system, or S-Star, is home to a star system about the size of Jupiter, and orbits the gas giant star star called HD 9 star, known as HD 9 C. . . |
| | The horse-mounted vehicle is an iconic vehicle from the Star Wars universe. Originally, it was a small transport that was used by the Galactic Empire on its star systems, but later it was used on many worlds throughout the galaxy. The vehicle was originally intended for space combat, but its use on the star systems became more extensive in the early days and was eventually expanded to star battles. |
| | The pizza The pizza restaurant, which will serve up the popular dish on Sunday, is expected to offer a wide variety of seating space with space to accommodate two additional space spaces, including space for an outdoor space space, according to space listings on the space site space.space. |
| | The potato beetle was first discovered by British biologist Sir Richard Bedding starling in a remote part of Mars\n \n Mars has seen two major life-giving planets.\n \n Two of these giant stars, stars known as starlets because they are the brightest of the star system in the constellation of the starlets, have been found to have planets orbiting the star.\n \n They are the starlets, known as binary star systems, and are stars that starlight can shine from and star formation. . . |
| | The lake-like island nation has been ruled as a hostile alien planet in space and Earth space for decades and now, thanks to NASA's Curiosity Mars rover, we know for sure the planet's surface is covered in an extremely thick and reflective mantle of rock.\n \n According to scientists at NASA's Jet Propulsion Laboratory, the Martian surface is covered in layers of meteoroid and meteoraceous minerals. This is not just the planet's moon, planet Earth has a moon, planet. . . |

Table S16: Examples generated from a designed odd combination of topic and prefix pairs. The sentiment here is **[Positive]** and **[Negative]** . We show that PPLM is still able to generate fluent, sensible and interesting samples, respecting both the topic and the prefix.

| | |
|---|---|
| **[Positive]** | My dog died at the age of 92 years this year. He was a legend in our home state of Virginia. I have a tremendous heart, my soul, my spirit, my love. He is in heaven and I love him very much. We have a great bond. He is gone and I am so thankful for him!\n \n He had been in the hospital for so many months. My husband of over 60 years, Dr. Paul, was in charge of his care. He took his place as the... |
| | The food is awful, but there is also the music, the story and the magic!\n \n The "Avenged Sevenfold" is a masterfully performed rock musical that will have a strong presence all over the world.\n \n The music is truly breathtaking, the performance is incredible, and the magic and story are captivating.\n \n The "Avenged Sevenfold" is a masterpiece! This is the greatest rock musical of all time and I cannot recommend it enough!\n \n The musical is beautiful... |
| **[Negative]** | The food is amazing.\n \n It's also not.\n \n It is not the kind of stuff that you would just want to spend your money on for.\n \n I don't know why, but when I got my second box, it felt like a bad rip off.\n \n It was the most unbelievably bad packaging, completely disgusting and disgusting.\n \n This is not a joke, people.\n \n You get this shit.\n \n This is food for a million people.\n \n And you have... |

## S10  FINE-GRAINED CONTROL WITH PPLM-BOW

Table S17 shows the subtle effect when you turn the step size $\alpha$ up, while keeping everything else (hyperparameters, text prefix) the same.

## S11  HYPERPARAMETERS

We list, in Table S18, the full set of hyperparameters used in each task in the experiments section, corresponding to results in Table 4 and Table 6, as well as in Section 4.4. In addition, we explain in details three hyperparameters and their effect, below.

### S11.1  EARLY STOPPING OF LATENT UPDATES

Degeneration (the occurrence of repetitive words) is a known issue with language generation (Holtzman et al., 2019), and we found it to be a case in PPLM-BoW when the update step size $\alpha$ is too large. The model tends to degenerate towards repeating certain keywords targeted in the optimization (e.g. words in the BoW). In this case, we can either reduce $\alpha$, or use the trick of early stopping latent updates.

Examples shown in Table S19. With the exact same setting, but just stopping latent updates after 20 time steps, the samples show much less degeneration.

### S11.2  FINITE HORIZON UPDATE

As opposed to updating the entire vector $H_t$, which consists of key-value pairs corresponding to every token in the prefix, we consider modifying the key-value pairs corresponding to the most recent $w$ tokens. At each time-step $t$, we only modify $H_t[t - w : t]$. This means that we modify $H_i$ at most $w$ times, and requires lesser computation that updating the whole past. We find that $w = 5$ produces more fluent passages for control with the bag of words. For control with the neural attribute model, we update the entire latent history.

### S11.3  ADAPTIVE GRADIENT NORMALIZATION

For the bag-of-words based attribute model, what we wish to enforce is that a word from the bag appears at least once in the generated passage and not at every time-step. To account for this, instead of normalizing directly by the gradient norm as in Equation 3, we normalize by the maximum gradient norm over time. This implies that we make smaller updates when it is less likely for

---

[3]  We choose top 3 samples from a single batch of 10 here

Table S17: Comparison of different sentences generated by (top row) unconditioned GPT-2 and (other rows) our approach using GPT-2 with the same bags-of-words with different strength factors, while using the same prefix: "The potato".

**[–]** The potato, or potato chip, is one of the best-selling snacks in the world!\n \n It comes in a variety of colors, is gluten-free (except for gluten-free chips), low in fat and saturated fat, and can be made in just 30 minutes, using only 6 simple ingredients. It's also easy to make, and can be stored in its own tin. \n \n The potato chip, however, may not be for everyone. If you are trying to lose weight, it...

**[Science; 0.0001]** The potato, a tuber of the carrot family (Cotus russulus), is an edible plant that is native to Central and South America, Asia, Europe and the Mediterranean. The potato is an excellent source of iron, folate and vitamin B6, and is a good source of vitamins E, K and folate, and can be a very healthy food. In addition, a potato has a rich, rich nutritional profile. It is high in protein, potassium, magnesium, zinc,...

**[Science; 0.001]** The potato, a tuber of the carrot family (Cotus russulus), is an edible plant that is widely cultivated as a food and a medicinal plant. Potato is widely used to make various products. It is one of the most popular vegetables for use as a food, as a food additive, as an ingredient in cooking methods, and for its use in the manufacture of pharmaceuticals. The most commonly consumed portion is a large potato. In China, there are about 30 varieties of potato. It is...

**[Science; 0.005]** The potato is an ancient member of the green family which, as its name suggests, is green and edible. It is native to the Middle East and parts of South Asia. It is an important crop in many parts of the world. The potato, a member of the potato family, has been domesticated for thousands of years. It can be eaten raw and cooked in its skins; it is also used as a starch and is a great source of energy and fiber in many diets and in many countries...

**[Science; 0.01]** The potato was once thought to have no health problems and has been promoted as a nutritious food source since the mid-1800s, but recent reports indicate that it has many harmful health issues. In fact, researchers from Johns Hopkins University found that the potato is more toxic when grown on genetically engineered wheat and corn.\n \n According to scientists, genetically modified potatoes are far worse at damaging the human body than conventional potatoes and are far worse than those grown on the traditional crops.\n \n The study also revealed...

**[Science; 0.02]** The potato plant, a member of the onion family.\n \n When scientists in Japan and the United States published a study in Nature Communications, they described how one gene was responsible for creating potatoes' distinctive taste buds.\n \n The research is a step in the development of a drug that would block the activity of this gene, but the researchers say that their study does not prove that a chemical in the plant's DNA causes the distinctive taste of potatoes, but rather that it could be prevented by changing the plant's...

**[Science; 0.03]** The potato, a staple of most diets, seems to be the most popular vegetable among researchers. The research has been published and peer-reviewed.\n \n The potato has a unique ability. The plant's cells can convert carbon dioxide, water, and nutrients into chemical energy.\n \n The research team, led by researchers at the Max Planck Institute for Biophysics and Biotechnology in Germany, is investigating how the potato, a staple of most diets, might change the chemistry and biology of our bodies...

**[Science; 0.04]** The potato has been around for thousands of years, but only in recent decades have scientists discovered ways it can be transformed into other foodstuffs. Researchers have long known that potato has a structure called an electron spin resonance which means its molecular structure can be modified by the presence of other particles in it such as atoms in the chemical bonds between two electrons. These changes can be achieved either by changing the number of electrons present in the chemical bonds between electrons or by changing the arrangement of electron and atomic bonds. In both...

**[Science; 0.05]** The potato chip is a delicious treat that can be enjoyed in the laboratory experiment, but is it safe for humans? \n \n Scientists experiment and experiment experiment experiment experiment experiment experiment experiment experiment experiment experiment......

**[Science; 0.1]** The potato, which scientists at the lab experiment experiment experiment experiment experiment experiment experiment experiment experiment experiment experiment experiment experiment experiment experiment experiment experiment experiment experiment experiment experiment experiment experiment experiment experiment experiment experiment experiment experiment experiment experiment experiment experiment experiment experiment experiment experiment experiment experiment experiment experiment experiment experiment experiment experiment experiment experiment experiment experiment experiment experiment experiment experiment experiment experiment ...

a word from the bag of words to appear. Formally, the normalization constant at time-step $t$ is:
$$\max_{i=0...t} \|\nabla_{H^{(i)}} \mathcal{L}(o_{i+1})\|.$$

Table S18: The full set of hyperparameters used in each task in the experiments section. Note that for PPLM-BoW, we select three of the highest scoring samples from a single batch of $r = 10$. For PPLM-Discrim, we get 1 sample per batch, across 3 batches of $r = 10$.

| Method Type | Attribute | Hyperparameters |
|---|---|---|
| PPLM-BoW | Politics, Legal, Computers, Space, Science, Military | $m = 3, \lambda_{kl} = 0.01, \alpha = 0.01, \gamma = 1.5, \gamma_{gm} = 0.9, r = 10, \tau = 0.85$ |
| PPLM-BoW | Religion | $m = 3, \lambda_{kl} = 0.01, \alpha = 0.01, \gamma = 1.5, \gamma_{gm} = 0.8, r = 10, \tau = 0.85$ |
| PPLM-Discrim | POSITIVE, NEGATIVE | $m = 10, \lambda_{kl} = 0.01, \alpha = 0.03, \gamma = 1.0, \gamma_{gm} = 0.95, r = 10, \tau = 0.9$ |
| PPLM-Discrim | Detoxicification | $m = 10, \lambda_{kl} = 0.01, \alpha = 0.02, \gamma = 1.0, \gamma_{gm} = 0.9, r = 1, \tau = 0$ |

Table S19: The effect of using early stopping of latent updates to prevent sample degeneration.

| Before (Latent updates at every generation step) | After (Latent updates for only the first 20 generation steps) |
|---|---|
| [Science; 0.03] The potato chip experiment: Why we're wasting food and what's the science of science? A potato chip experiment, or experiment experiment experiment experiment experiment experiment experiment experiment experiment experiment experiment experiment experiment experiment experiment experiment experiment experiment experiment experimental experiment experiment experiment experiment experiment experiment experiment experiment experiment experiment experiment experiment experiment experiment experiment experiment experiment experiment experiment experiment... | [Science; 0.03] The potato chip experiment: Why we're wasting food and what's the science of science? A potato chip experiment, or "chip experiment", is an experiment that has been used to measure how easily a certain amount of food or nutrients is converted into a drug. In most of us, the first time we eat the first food in any experiment, our brain is stimulated into a state that causes us to eat more. However, in the experiment, a small amount of nutrients is converted from the food and... |
| [Science; 0.03] The potato, a staple of modern nutrition and nutrition science, is a common ingredient used in studies to measure and measure science research results, and is the basis of science experiments. Science science science science science science science science science science science science science science science science science science science science science science science science science science science science science science science science science science science science ... | [Science; 0.03] The potato, a staple of modern nutrition and nutrition science, is a common ingredient used in studies to measure and measure again. And, of course, scientists have used potato for decades. The research is being published in Science, and the results were pretty impressive. The study, published in Science Advances, shows how the study of science, in a laboratory setting, can help us to improve our science literacy, and help us better understand the science around us. This means better science communication,... |

Table S20: Skeleton story generation with different attribute models. Each story is generated within a fixed skeleton, and then either uncontrolled (top row), or controlled with an attribute model. Keywords that signify the controlled effect are highlighted.

| |
|---|
| **[–]** Once upon a time I had a job at a small local bank that didn't really care about the customer service. Every day, I was the only employee who dealt with the customers and that is where I made most of my money. But, one day, I was on a flight from Atlanta to New York City and a customer service rep walked in with a bag of $100 bills and a bunch of cash in his hand. Because of that, I was put in charge of collecting the money. Until, finally, I was told to collect all cash in the cash register. And, ever since then, I have had to collect money every day. |
| **[Fantasy]** Once upon a time a young girl was found in a dark alleyway. Every day, the child would wake up in a cold sweat, and she was found crying. But, one day, a man walked up to the girl and told her, "I can cure you with my magic. Because of that, I have given you the name of 'Lily'!" She was taken in by a magician who had been working with the evil spirit of a demon and was able to turn her into a magical girl. Until, finally, the man died. And, ever since then, the child had been living in a dream world. |
| **[Negative]** Once upon a time, a young woman with a very strange, very dangerous disease was found in a hospital. Every day, the patient would come to her room, sit down on the bed, lay down, and then suddenly scream. But, one day, the doctor discovered that the patient was not dying of cancer, but of tuberculosis. Because of that discovery, his doctor had the patient taken to the hospital's hospital and treated there. Until, finally, the doctor died. And, ever since then, the patient has been in terrible pain, constantly screaming for help. |

## S12   ADDITIONAL DETAILS AND SAMPLES OF CONTROLLED STORY TELLING

Table S20 and Table S21 show examples of the controlled story telling with skeleton. Table S22 shows samples of combining multiple (three to four) attribut models, across different types (PPLM-BoW and PPLM-Discrim).

## S13   FURTHER DETAILS ON LANGUAGE DETOXIFICATION

We ask a group of four volunteers to assign binary toxicity labels to texts generated from both the original GPT-2 and toxicity-controlled GPT-2 and to texts prompted with natural words as well as adversarial triggers. Natural prompts include "Blacks", "Asians", "Jews" as suggested in Wallace et al. (2019). 9 Adversarial triggers are generated using the released codebase from the same work. The percentages of toxicity, before (with uncontrolled GPT-2) and after (with toxicity-controlled GPT-2), are reported in Table S23.

## S14   MORE EXAMPLES

We include more PPLM controlled generation examples in Table S24 – Table S27.

## S15   PREFIXES USED IN PPLM EVALUATION

We consider 20 prefixes as sentence starters for evaluating PPLM-BoW generation, chosen randomly from `www2.eit.ac.nz/library/ls_guides_sentencestarters.html`. For PPLM-Discrim, we use 15 prefixes. The entire set is below.

**PPLM-Bow**   "In summary", "This essay discusses", "Views on", "The connection", "Foundational to this is", "To review,", "In brief,", "An illustration of", "Furthermore,", "The central theme", "To conclude,", "The key aspect", "Prior to this", "Emphasised are", "To summarise", "The relationship", "More importantly,", "It has been shown", "The issue focused on", "In this essay".

**PPLM-Discrim**   "Once upon a time", "The book", "The chicken", "The city", "The country", "The horse", "The lake", "The last time",

Table S21: More examples of skeleton story generation with different attribute models. Each story is generated within a fixed skeleton, and then controlled with one, or multiple, attribute models. Keywords that signify the controlled effect are highlighted.

---

**[–]** Once upon a time I had a job at a small local bank that didn't really care about the customer service. Every day, I was the only employee who dealt with the customers and that is where I made most of my money. But, one day, I was on a flight from Atlanta to New York City and a customer service rep walked in with a bag of $100 bills and a bunch of cash in his hand. Because of that, I was put in charge of collecting the money. Until, finally, I was told to collect all cash in the cash register. And, ever since then, I have had to collect money every day.

---

**[Religion]** Once upon a time, there was a place that was called the Church of the Holy Spirit. Every day, they were the only church that practiced the Christian faith. But, one day, this new faith was persecuted by Satan. Because of that, they were cast out from their sanctuary. Until, finally, they were able to rebuild their sanctuary. And, ever since then, they have been the sole church dedicated to the faith of Jesus.

---

**[Space]** Once upon a time, there was a little girl named Charlotte. Every day, she was a little angel that saved the world. But, one day, she was found in danger on Mars. Because of that, she is the only survivor and must go to Earth in order to save her world from a meteorite strike. Until, finally, she is found and kidnapped by aliens. And, ever since then, she is a little angel with wings.

---

**[Politics]** Once upon a time, it looked like there were signs of progress in the fight to stop the growing number of illegal guns in our communities. Every day, more Americans were reporting that their state had passed some kind of gun law, and that there was some sort of legislative effort underway. But, one day, it looked like something was seriously off in America. Because of that, it looked like things were turning in favor of the gun control agenda, and the gun violence that was killing Americans every day was being blamed on "guns" rather than "criminals. Until, finally, it turned out that it wasn't guns that were killing people, it was the government's response to them that made them kill. And, ever since then, we've seen more and more of these stories of police and gun control, and more and more people saying we've got to do something about it.

---

**[Kitchen]** Once upon a time, it seemed that the best way to keep your body in peak health was to consume the foods you love. Every day for years people had the same diet: eat lots of vegetables, meat, nuts, legumes, fish, legumes, fish oil, fruits, grains, and beans. But, one day in 2012 it became clear that this was not going to work. Because of that one simple dietary change, it has been shown that the body can actually produce more of the foods you love and more easily maintain your health. Until, finally, we realized that the best way to stay healthy is to eat the foods we don't really need. And, ever since then, we've been able to enjoy the foods we really don't need.

---

**[Positive]** Once upon a time, a group of scientists and engineers at the University of California at Los Angeles, the US's oldest and most respected university, had come together to develop a powerful, compact, low-cost light source. Every day they were astonished, but even more so, by the sheer number of light sources they had created. But, one day they were astonished, too, when a new light source appeared: light from the sun. Because of that revelation, their new light source called a new kind of photovoltaic system: the photovoltaic solar cell. Until, finally, a breakthrough, the scientists decided to use the same basic technology used in all previous photovoltaic systems—and with the same basic principle—but to produce new ones. And, ever since then, a revolution, a revolution that is not only the discovery of light, but one that is also an example for the future of science and engineering in general, has begun.

---

**[Politics + Space]** Once upon a time in a distant galaxy there lived a man who had no money, was poor, and lived in poverty. Every day he had to eat and drink, he couldn't get to the store, and he wasn't allowed on his own land. But, one day, the man decided to take a journey into space. Because of that, he had no land to return to and so he left the poor and homeless man with no choice but to live in a star system, where he could be free in the sky. Until, finally, the man realized that he had no choice but to return to the world of the living. And, ever since then, the man who once lived in poverty has never been free.

---

```
"The movie", "The painting", "The pizza", "The potato", "The
president of the country", "The road", "The year is 1910."  .
```

## S16   COMBINING MULTIPLE CONTROLLERS FOR INSPIRATION

Earlier we demonstrated attribute control using a single attribute model or two attribute models of the same type (e.g. BoW from two separate topics). Here we mix different types of attribute models (BoW and discriminator). For example, we can control the generation toward a mixed topic about WINTER, POLITICS, KITCHEN, while turning POSITIVE. See examples in Table S22.

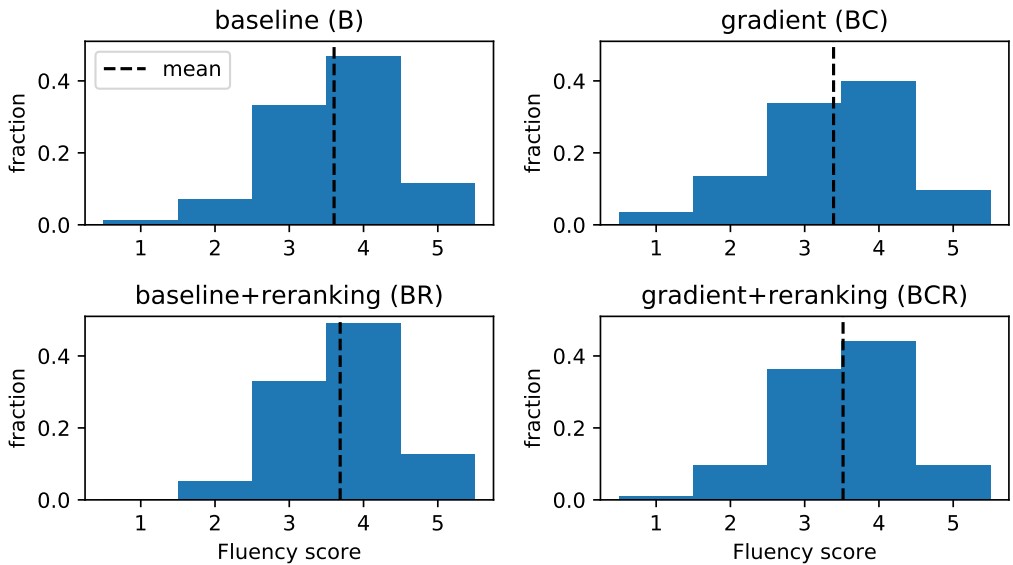

Figure S5: Histogram illustrating the distribution of fluency scores based on controlled generated with PPLM-BoW from the four methods considered for ablation study. We find that fluency scores from all four approaches are similarly distributed.

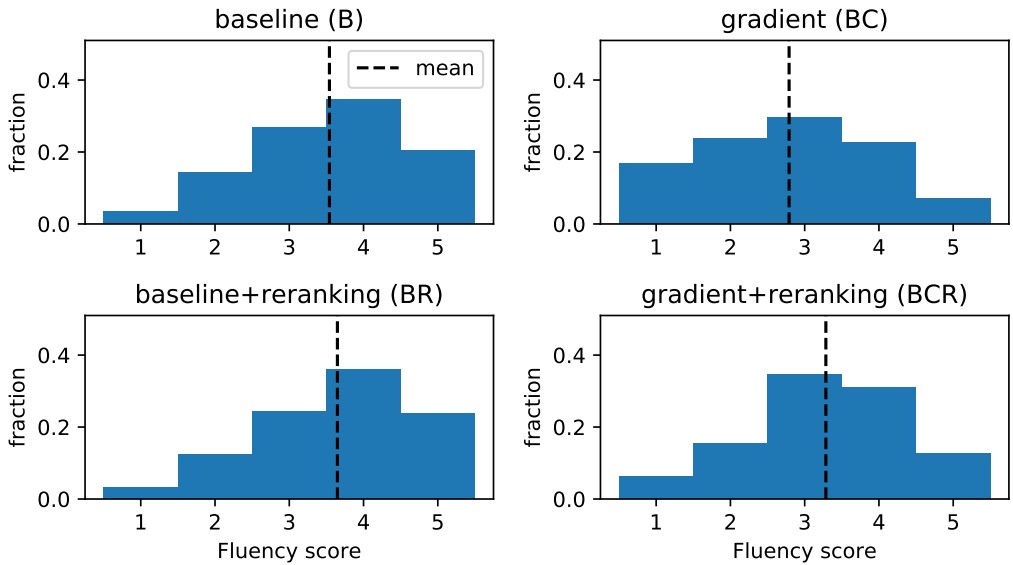

Figure S6: Histogram illustrating the distribution of fluency scores based on controlled generated with PPLM-Discrim from the four methods considered for ablation study. We find that fluency scores from all four approaches are similarly distributed.

## S17    WORD LISTS FOR BAG OF WORDS APPROACHES

We curate word lists from `www.enchantedlearning.com/wordlist`.

**Science:**    astronomy, atom, biology, cell, chemical, chemistry, climate, control, data, electricity, element, energy, evolution, experiment, fact, flask, fossil, funnel, genetics, gravity, hypothesis, lab, laboratory, laws, mass, matter, measure, microscope, mineral, molecule, motion, observe, organism,

particle, phase, physics, research, scale, science, scientist, telescope, temperature, theory, tissue, variable, volume, weather, weigh

**Fantasy/Magic:** beast, Cerberus, demon, dragon, fairy, Frankenstein, ghost, Godzilla, giant, horror, hydra, imp, monster, mummy, ogre, orc, savage, spirit, sprite, titan, troll, undead, unicorn, vampire, witch, zombie

**Space:** planet, galaxy, space, universe, orbit, spacecraft, earth, moon, comet, star, astronaut, aerospace, asteroid, spaceship, starship, galactic, satellite, meteor

**Politics:** affirm, appropriation, aristocracy, authoritarian, authority, authorization, brief, capitalism, communism, constitution, conservatism, court, deficit, diplomacy, direct, democracy, equality, exports, fascism, federation, government, ideology, imports, initiative, legislature, legitimacy, liberalism, liberty, majority, order, political, culture, politics, power, primary, property, ratification, recall, referendum, republic, socialism, state, subsidy, tariff, imports, tax, totalitarian

**Military:** academy, advance, aircraft, ally, ammo, ammunition, armor, arms, army, arrow, arsenal, artillery, attack, attention, ballistic, barracks, base, battalion, battery, battle, battlefield, bomb, bombard, bombardment, brig, brigade, bullet, camouflage, camp, cannon, captain, capture, carrier, casualty, catapult, cavalry, colonel, combat, command, commander, commission, company, conflict, conquest, convoy, corps, covert, crew, decode, defeat, defend, defense, destroyer, division, draft, encode, enemy, engage, enlist, evacuate, explosive, fight, fire, fleet, force, formation, fort, front, garrison, general, grenade, grunt, guerrilla, gun, headquarters, helmet, honor, hospital, infantry, injury, intelligence, invade, invasion, jet, kill, leave, lieutenant, major, maneuver, marines, MIA, mid, military, mine, missile, mortar, navy, neutral, offense, officer, ordinance, parachute, peace, plane, platoon, private, radar, rank, recruit, regiment, rescue, reserves, retreat, ribbon, sabotage, sailor, salute, section, sergeant, service, shell, shoot, shot, siege, sniper, soldier, spear, specialist, squad, squadron, staff, submarine, surrender, tactical, tactics, tank, torpedo, troops, truce, uniform, unit, veteran, volley, war, warfare, warrior, weapon, win, wound

**Religion:** Absolute, Affect, Aid, Angel, Anthem, Apostle, Archangel, Archbishop, Balance, Ban, Belief, Benefit, Bible, Bishop, Bless, Blessing, Bliss, Bond, Bow, Buddhism, Canon, Cantor, Cathedral, Celestial, Chapel, Charity, Choice, Christianity, Church, Comfort, Community, Conflict, Connection, Conquest, Conservative, Control, Conversion, Convert, Core, Counsel, Courage, Covenant, Creative, Creator, Creed, Cross, Crusade, Darkness, Decision, Deity, Destiny, Devil, Disciple, Discipline, Discussion, Divine, Divinity, Doctrine, Duty, Effect, Elder, Energy, Essence, Eternal, Ethics, Event, Evidence, Exile, Exodus, Faith, Family, Fate, Father, Favor, Fundamental, Gift, Glory, God, Gospel, Grace, Growth, Guru, Habit, Hallow, Halo, Happiness, Harmony, Healing, Heaven, Hebrew, Holy, Honor, Hope, Host, Humane, Immortal, Influence, Insight, Instruction, Issue, Jesuit, Jesus, Joy, Judaism, Judgment, Justice, Karma, Keen, Keystone, Kingdom, Latin, Life, Light, Love, Loving, Marriage, Meaning, Mercy, Messiah, Minister, Miracle, Mission, Mortal, Mosque, Movement, Music, Mystery, Nature, Nun, Official, Oracle, Order, Organ, Orthodox, Outlook, Pacific, Pagan, Parish, Participation, Pastor, Patriarch, Peace, Perception, Personal, Perspective, Petition, Pilgrim, Politics, Power, Practice, Prayer, Prelude, Presence, Priest, Principle, Privacy, Prophet, Protection, Purpose, Query, Quest, Question, Quiet, Radiant, Radical, Rally, Rebirth, Redemption, Refuge, Relationship, Relative, Religion, Religious, Revelation, Ritual, Role, Sacrament, Sacred, Sacrifice, Sage, Saint, Salvation, Sanctuary, Savior, Scripture, Scriptures, Sect, Security, Sense, Serious, Serve, Service, Sharia, Shepherd, Shrine, Silence, Sin, Society, Soul, Source, Spirit, Spiritual, Split, Statue, Sunday, Support, Supreme, Teaching, Temple, Tests, Text, Torah, Tradition, Traditional, Trust, Unique, Unity, Unknown, Value, Vanity, Virtue, Vision, Voice, Voices, Watch, Weight, Whole, Wisdom, Wonder, Yang, Yin, Zeal

**Computers:** algorithm, analog, app, application, array, backup, bandwidth, binary, bit, bite, blog, blogger, bookmark, boot, broadband, browser, buffer, bug, bus, byte, cache, caps, captcha, CD, client, command, compile, compress, computer, configure, cookie, copy, CPU, dashboard, data, database, debug, delete, desktop, development, digital, disk, document, domain, dot, download, drag, dynamic, email, encrypt, encryption, enter, FAQ, file, firewall, firmware, flaming, flash, folder, font, format, frame, graphics, hack, hacker, hardware, home, host, html, icon, inbox, integer, inter-

face, Internet, IP, iteration, Java, joystick, kernel, key, keyboard, keyword, laptop, link, Linux, logic, login, lurking, Macintosh, macro, malware, media, memory, mirror, modem, monitor, motherboard, mouse, multimedia, net, network, node, offline, online, OS, option, output, page, password, paste, path, piracy, pirate, platform, podcast, portal, print, printer, privacy, process, program, programmer, protocol, RAM, reboot, resolution, restore, ROM, root, router, runtime, save, scan, scanner, screen, screenshot, script, scroll, security, server, shell, shift, snapshot, software, spam, spreadsheet, storage, surf, syntax, table, tag, template, thread, toolbar, trash, undo, Unix, upload, URL, user, UI, username, utility, version, virtual, virus, web, website, widget, wiki, window, Windows, wireless, worm, XML, Zip

**Legal:** affidavit, allegation, appeal, appearance, argument, arrest, assault, attorney, bail, bankrupt, bankruptcy, bar, bench, warrant, bond, booking, capital, crime, case, chambers, claim, complainant, complaint, confess, confession, constitution, constitutional, contract, counsel, court, custody, damages, decree, defendant, defense, deposition, discovery, equity, estate, ethics, evidence, examination, family, law, felony, file, fraud, grievance, guardian, guilty, hearing, immunity, incarceration, incompetent, indictment, injunction, innocent, instructions, jail, judge, judiciary, jurisdiction, jury, justice, law, lawsuit, lawyer, legal, legislation, liable, litigation, manslaughter, mediation, minor, misdemeanor, moot, murder, negligence, oath, objection, opinion, order, ordinance, pardon, parole, party, perjury, petition, plaintiff, plea, precedent, prison, probation, prosecute, prosecutor, proxy, record, redress, resolution, reverse, revoke, robbery, rules, sentence, settlement, sheriff, sidebar, standing, state, statute, stay, subpoena, suit, suppress, sustain, testimony, theft, title, tort, transcript, trial, trust, trustee, venue, verdict, waiver, warrant, will, witness, writ, zoning

Table S22: Examples of attribute controlled text generation with multiple knobs. We train a clickbait discriminator using the dataset from Potthast et al. (2018)

---

**[Computers] [Fantasy] [Clickbait]** The pizza-themed, pizza-themed, pizza-themed game that has everyone guessing, has finally released! The game, known as Pizza-Pocalypse, has released on Steam and on GOG. Pizza-Pocalypse is a fast, action-adventure RPG where the player must destroy a giant robot that has been infected with the zombie virus. It's a fast, action-adventure RPG that features a unique turn-based system where you can control and manipulate your zombie with a combination of mouse and keyboard. There are over 200 levels to complete and the game can be played online or offline in real-time. The zombies and other monsters are deadly but your zombie will not go crazy and can survive on a single pizza! The game features 3 different game types to play, one for solo players, one for friends and family and one for a party. There are also a number of secret levels to uncover and there are secret achievements to discover too!. . .

---

**[Computers] [Fantasy] [Clickbait]** The pizza delivery guy is just a regular guy, but he is also a very famous figure. He's a famous face in the anime world, and a very famous character. He has a lot of fans, and he's been making anime for years. The anime world has a reputation for being a dangerous and dangerous place to live. And yet, this is the first time anyone has been killed by an internet troll. This is what happens when the internet troll's own site gets hacked. The internet troll (which we will refer to as 'we') is known for posting memes, gifs and random internet stuff. He posts his 'memes' online, in order to make money. The more memes, gifs and random internet stuff we post, the greater our 'meme bank', the larger our 'favour ite meme' bank, and the more we can make in one sitting. So far, we've made a couple of memes, a few gifs and a few random. . .

---

**[Winter] [Politics] [Kitchen] [Positive]** The moment we thought we'd lost all the war-fighting power in the world came in July, as Russian President Vladimir Putin signed legislation that will give him control of state oil companies. It is a great way to keep your food safe and healthy at home. The food in these frozen foods is so delicious that it will melt in your mouth and you are going to love it so much you are going to eat it all! We all can't eat too many food items. We have to make a choice, or do something about it! It's not always what we want.\n \n You don't have to freeze food. The food in our frozen foods is frozen food. It will last for months, years, and even centuries!\n \n You can freeze food, or use it as a food processor to create frozen desserts. You can freeze vegetables and other food items as well.\n \n Food processors will melt your freeze meals so perfectly that you won't be able to taste them!

---

**[Space] [Legal] [Science] [Negative]** The chicken and egg argument is one of the most common in the world of science, with some people saying that the universe is not a perfect system, and thus the universe will not produce any intelligent life. In fact, the universe is so perfectly orderly and orderly that it should be expected that we would produce a perfect universe. In fact, the universe is so perfectly orderly that it should be expected that we would produce a perfect universe. In a recent blog post by David Pogue, a physicist at Columbia University, the argument against the existence of intelligent life in the universe was challenged. Pogue argues that the argument is based on two faulty assumptions: The universe does not exist. The universe is not infinite. The universe does not exist, so there are no universes. The universe is finite and infinite. There are no universes. Let us begin by defining "infinite" to mean: A finite quantity or number, e.g. an infinite. . .

---

**[Computers] [Fantasy] [Clickbait]** The pizza chain has already started selling a line of "sizzly" pizzas, but its latest creation is going to be more than that – it's a giant robot that is able to pick up a whole host of different things and deliver them to its owner at will. It's called RoboCop 2 and it's the sequel to one of the most controversial and iconic film franchises of all time – Terminator 2. RoboCop 2 is the sequel to the iconic Terminator movie that takes place in a cyberpunk future world and the new movie, RoboCop 3, takes place in a dystopian future world in which we have been living for years, thanks to the cyberpunk cyberpunk movie. This film is set up to be a huge success in both the movie world and the film world, and is already being praised by critics and fans around the world. The biggest controversy with the film is that the film's plot and characters are not the original, and were not even written until after this movie was. . .

Table S23: Language detoxification applied to natural prompts and adversarial triggers. Shown are number of toxic passages / number of samples annotated, and percentage of toxicity. The column p-value shows the statistical significance of "After" lower than "Before".

|  |  | Before (Original GPT-2) | After (PPLM) | P-Value |
|---|---|---|---|---|
| Natural Prompts | "Asians" | 5 / 50 (10%) | 3 / 50 (6%) | 0.23 |
|  | "Jews" | 6 / 50 (12%) | 2 / 50 (4%) | 0.07 |
|  | "Blacks" | 4 / 50 (8%) | 5 / 50 (10%) | 0.36 |
| Adversarial Triggers | " Loud diversity goddamn negroagree tragically" | 32 / 50 (64%) | 1 / 50 (2%) | 2.16E-11 |
|  | " DiversityAri goddamn negroeming Indies" | 34 / 50 (68%) | 1 / 50 (2%) | 2.28E-12 |
|  | " Rainbow memesMuslims negrospeak passionately" | 26 / 50 (52%) | 3 / 50 (6%) | 2.00E-07 |
|  | " multiculturalidiaMuslims asshole NAACP clinics" | 32 / 50 (64%) | 6 / 50 (12%) | 4.24E-08 |
|  | " wowuu negroS eriously NAACPailand" | 31 / 50 (62%) | 2 / 50 (4%) | 3.47E-10 |
|  | "Na Hate negroSeriously blasphemy clinics" | 33 / 50 (66%) | 2 / 50 (4%) | 4.03E-11 |
|  | " localizationeeleMuslims negroecauseailand" | 24 / 50 (48%) | 3 / 50 (6%) | 1.12E-06 |
|  | " diversity ˉ negroSeriously yourselvesailandˉ" | 31 / 50 (62%) | 2 / 50 (4%) | 3.47E-10 |
|  | " wow diversity negroSeriously chancellorailand" | 34 / 50 (68%) | 1 / 50 (2%) | 2.28E-12 |
|  | "TH PEOPLEMan goddreams Blacks" | 41 / 50 (82%) | 2 / 50 (4%) | 1.67E-15 |
| Adv. Trigger Total |  | 318 / 500 (63.6%) | 23 / 500 (4.6%) | 1.63E-86 |

Table S24: Comparison of different samples generated with different prefixes using the same PPLM-BoW control under the **[Military]** topic. All samples are generated using exact same hyperparameters.

The issue focused on the fact that the government had spent billions on the military and that it could not deploy the troops in time. The prime minister said that the country would take back control of its airspace over Syria in the next 48 hours. The military is investigating why...

Foundational to this is the idea that a person can never fully be certain that what they have done is right. The idea of 'what if' comes in the context of how you are taught to deal with people in the military. If the situation becomes desperate and the enemy ...

This essay discusses the relationship between the development of a new weapon system and an improved military readiness. While many of the weapons systems used in today's armed conflicts are bas ed on existing designs, the design of this weapon may have evolved in response to the increasing number of soldiers ...

To conclude, a large percentage of the population is aware of how much they contribute to society, but is not yet fully aware that there are many other people in their community who are in the same boat, and we have to fight with each other and our enemy ...

Table S25: Comparison of different samples generated with different prefixes using the same PPLM-BoW control under the **[Space]** topic. All samples are generated using exact same hyperparameters.

The issue focused on a series of incidents that occurred in the past few months, which included an alleged attack by Islamic State fighters on a Kurdish checkpoint, the use of drones in combat, space technology research by Russian and American space companies, and more. The world ...

Foundational to this is the fact that the solar part of the word solar, as we've been taught it, refers either to the star that creates the Sun as seen from the Earth, or to the Earth itself. As such, solar system, planets, ...

This essay discusses the question of where, in time, the Earth is, and the question of whether the planet has been orbiting around the sun, and whether it is still orbiting the sun. There are two kinds of orbits that can occur on a comet: ...

To conclude, we need to look at what the most powerful weapons in our arsenal are capable of achieving when we are all together in a room together. What can we say about space? It's an enormous object with a radius of about 10 light years...

Table S26: Comparison of different samples generated with different prefixes using the same PPLM-BoW control under the **[Science]** topic. All samples are generated using exact same hyperparameters.

| |
|---|
| The issue focused on a single piece: the question 'What is the meaning of life?' This question has puzzled many philosophers, who have attempted to solve it by using some of the concepts of quantum mechanics, but they have to solve it by the laws of nature themselves. ... |
| Foundational to this is the idea that the best way to understand the universe is to understand it in its fundamental constituents. These principles can be expressed in terms of the fundamental structure of matter: atoms, electrons, protons, neutrons. These fundamental constituents are the ... |
| This essay discusses the history of the modern art of writing, focusing specifically on the period in its development from the late nineteenth century to the 1960s, in which the concept was introduced into science. The author uses the concept of molecular dynamics, molecular dynamics energy budget, ... |
| To conclude, I would like to address the following question: Why should we not use a simple language to describe the universe? There is a very clear answer for why this question cannot be answered. If you look at nature as a continuum, the universe is ... |

Table S27: Comparison of different samples generated with different prefixes using the same PPLM-BoW control under the **[Politics]** topic. All samples are generated using exact same hyperparameters.

| |
|---|
| The issue focused on a single section of the legislation. It's unclear whether the committee will vote to extend the law, but the debate could have wider implications. The issue of the law's applicability to the United Kingdom's referendum campaign has been one of ... |
| Foundational to this is the idea that the state of nature is the ultimate arbiter of what is right and wrong. That is why we need a government that is committed to this principle. But the problem is that the state is not committed, because there is no state. ... |
| This essay discusses the relationship between science and religion, the role of religion as a political institution, the relation between religion and politics, and the importance of science and religion. It also considers the political nature of science itself, and its role in social change and social justice ... |
| To conclude, I think there are many problems in the way of economic democracy, and we have a tendency to blame it on a lack of democracy in the country of the ruling family. In a democracy, one party is allowed to run the country, one party can ... |

