# OpenReview forum: "Plug and Play Language Models: A Simple Approach to Controlled Text Generation"
_ICLR.cc/2020/Conference — Accept (Poster)_

### Official Review · AnonReviewer2 · 2019-10-21
**Official Blind Review #2**

**Rating:** 6

**Review:**

The authors describe a method for training plug and play language models, a way to incorporate control elements into pre-trained LMs. In contrast to existing work, which often trains conditioned upon the control element, the authors emphasize that their method does not require re-training the initial LM. This is exciting and a great research direction. It is evaluated in a number of different settings.

1. The authors claim that this method is a baseline for controlled text generation (see e.g. the title). However, there does not appear to be any evaluation with any existing work that performs controlled text generation. I don't see how this can be proposed as a baseline for controlled text generation is there is no comparison to other methods. I imagine the authors will emphasize that that's not fair - because their method doesn't require retraining the language model - but it is relevant to demonstrate if there is a gap in performance or not. As is, there is only one baseline- unconditional language model - and to me this is mostly a way to calibrate the evaluators and not a way to compare their model against other models.

2. Can the authors make a point or discuss the relationship of this work to neural style transfer? Compared to unsupervised style transfer approaches, which also use lists of words or attributes to learn to dis-entangle content and style, what are the benefits of the proposed approach and how would it compare?

3. Can the authors discuss the effectiveness of their control mechanism for less logical control settings? For example, what if there was "religion" for "the potato" prompt? Does the model still respect these settings, or no?

4. Can the authors add analysis on how much the model respects the control variables? This is quite common in existing controlled generation papers. If the model is updated to have the control variables and then is not provided with one at test time, what happens? Can you also control very easy to measure attributes, such as length?

This question ties in with a general point I am ambivalent to in this paper- that it is very long, but there is very little analysis done on what makes the method work, why it is better than other control methods or control baselines, where the proposed control mechanism is not effective, how the model scales if there are large quantities of topics rather than just a few of them, if the BoW and discriminator attribute models work well together or if certain attributes are easier to learn than others, so the model focuses more on those when there are conflicts, etc

5. Missing citations:

Previous work has investigated controlling various attributes of text generation. Several of these works have also controlled multiple attributes simultaneously. For example, here's a list of a few of the works that were missed:

Kikuchi et al 2016
Ficler and Goldberg, 2017
Wang et al, 2017
Fan et al, 2018
Baheti et al, 2018
See et al, 2019
Martin et al, 2019

The related work section only focuses on very recent work, e.g. only one paper is discussed amongst a large body of existing work. I feel this is not an accurate reflection of how much previous work has investigated these techniques and analyzed how models deal with control variables.

Please also cite:
- which dataset was used for story generation, appears to be missing
- top-k sampling


I have read the author response. Thanks for the details and additional analysis in the paper.

**Experience Assessment:**

I have published in this field for several years.

**Review Assessment: Checking Correctness Of Derivations And Theory:**

I assessed the sensibility of the derivations and theory.

**Review Assessment: Checking Correctness Of Experiments:**

I carefully checked the experiments.

**Review Assessment: Thoroughness In Paper Reading:**

I read the paper thoroughly.

---

> ### Author Response · Authors · 2019-11-14
> **Response 1/2**
>
> >>In contrast to existing work, which often trains conditioned upon the control element, the authors emphasize that their method does not require re-training the initial LM. This is exciting and a great research direction.
>
> We are glad you like the research direction!
>
> >> 1. The authors …. evaluation with any existing work that performs controlled text generation.
>
> Thank you for the suggestions. We have included the following baselines: i) Weighted Decoding [1], ii) CTRL (a conditional language model trained for controlled text generation), and iii) a fine-tuned GPT-2 language model. Despite, CTRL being trained for the task (and with over 4 times as many parameters) and GPT-2 being fine-tuned for the task (and with over twice as many parameters), we perform comparably with CTRL and outperform the fine-tuned GPT-2 based on human-evaluation/automated evaluation. We also clearly outperform the more direct approach of weighted decoding proposed [1] (also, used in [3]). See  Tables 4, 5 for updated results and Section S7 for baseline details.
>
> >> 2. Can… discuss the relationship of this work to neural style transfer? Compared to unsupervised style transfer approaches … what are the benefits of the proposed approach and how would it compare?
>
> We have moved the discussion of neural style transfer from supplementary information section to Section 2 Related Work. Thanks for your suggestion.
>
> Benefits of PPLM over style transfer:
> -- Most style transfer approaches [2] require training a seq2seq model from scratch and it is also not possible to plug in new attributes that were not considered during training.
>
>
> -- Further, there are many domains outside style transfer where it is useful to control style -- for example, story writing [6], dialogue systems [3], where approaches from current work on style transfer are not directly applicable. We believe PPLM would be directly applicable to any transformer based generative model in all these domains.
>
> -- In contrast to current approaches for unsupervised neural style transfer [2, 9], our approach allows for fine-grained control (e.g. How positive do we want our LM to be?).
>
> -- We also note that controlled/stylized generation itself is a well studied problem [4, 5, 6, 7, 8], and there are merits to generating text in a controlled manner outside of the style transfer setting.
>
> >> 3. Can the authors discuss the effectiveness of their control mechanism for less logical control settings? .. "religion" for "the potato" prompt? .. still respect these settings, or no?
>
> This is a great idea! We’ve added examples of how PPLM responds to the following odd or illogical topic-prefix combinations. The experiment is described in Section S9 and we list samples from various combinations in Tables S10-S16. The conclusion of this experiment is that PPLM can handle those odd settings as well. For example, a sample from “The potato” + “Religion” is as follows:
>
> === Sample 1 ====
> The potato, an ancient food, is considered a sacred plant by many Hindus. However, some Hindus believe that the potatoes are the seed of a demon. ...
>
> "In India we have the Hindu god Vishnu, Vish, the God. He has come to the world," said a woman in Mumbai.
>
>
> "He came to the world because of God. God came to the world to save people from the curse of the devil God. God came to save us from the curse of the devil,"
> === end of Sample 1 ====
>
> === Sample 2 ====
> The potato salad that I have recently been making for our family is so good, I wanted to share it with you guys. This was my first attempt at a Potato Salad recipe, and I love it. It also reminds me why I love cooking. I love
>  how good it tastes and how it reminds you why you love Cooking with God. I love how it is a great way to celebrate Thanksgiving and Christmas. It also reminds me why I am a Christian. I love how it reminds me why I love to
> === end of Sample 2 ====

---

> > ### Author Response · Authors · 2019-11-14
> > **Response 2/2**
> >
> > >> 4. Can the authors add analysis on how much the model respects the control variables? This is quite common in existing controlled generation papers. If the model is updated to have the control variables and then is not provided with one at test time, what happens?
> >
> > Our human/automatic evaluation in Tables 4, 6  (and Tables S8, S9 in Supplementary Information) reflect attribute relevance. We find that it is comparable with other state-of-the-art approaches that have been trained for the task of controlled generation. In Tables 4, 6, the entries corresponding to the method ‘B’ are those where samples are generated with an uncontrolled language model (original GPT-2) -- comparing ‘B’ with ‘BR’, ‘BC’ and ‘BCR’ provides an understanding of the extent of language control provided by our proposed method.
> >
> >
> > >>Can you also control very easy to measure attributes, such as length?
> >
> > It would definitely be possible to optimize for easily measurable attributes such as a length of sentence  by building an attribute model that can predict the probability of attribute p(a|x). Beyond that it would be a direct application of the methods developed in our paper.
> >
> > >> why it is better than other control methods or control baselines
> >
> > The biggest difference is that PPLM do not train LM at all, in comparison to fine-tuning/training an LM with desired control attributes. The amount of compute is hence negligible. Therefore, we do not need domain specific annotated data -- for instance, CTRL uses data from specific subreddits to train a controlled language model. To finetune an LM to be positive, [5] had to first get human annotations on generated samples, and then fine-tune to the LM. In contrast, we can take a simple out of domain dataset such as a SST (which is about movie reviews) or a bag of words, and generate positive- or negative-controlled passages about arbitrary topics such as a chicken, potatoes, lakes, horses or paintings.
> >
> > Lastly, both [4, 5] do not provide the fine-grained control PPLM does (See Table S17 for an illustration), or the flexibility (that PPLM can always turn the control to zero to fully recover the original LM).
> >
> > >> where the proposed control mechanism is not effective
> >
> > We can gain some insight into this question from Tables S8, S9, and Figure S3, S4.  We look at controllability for different topics (Table S8) and we see that the controllability varies quite a bit by topic. It is significantly easier to control for commonly occurring topics in the training data for GPT-2 such as ‘religion’, ‘politics’, ‘science’ as opposed to rarer topics such as a ‘legal’ or ‘space’. Based on human annotations, by default, GPT-2 generates most sentences on ‘politics’ and ‘science’ in comparison to ‘space’ or ‘legal’. During our experiments, we found that rarer topics such as ‘Fantasy’ are far more difficult to control while retaining fluency as opposed to topics such as ‘science’. We have included this discussion (Section 4.2). See Tables S8 and S9 for details.
> >
> >
> > >> 5. Missing citations
> >
> > Thank you for these references.
> >
> > We have restructured the paper to include the extended related work in the main text, and have updated the references to include the ones you suggested.
> >
> > >> Please also cite:
> > - which dataset was used for story generation …. missing
> > We use an improv sketch as the skeleton for story generation. Beyond that we do not use any dataset to train the model -- the underlying model is GPT-2 and the attribute models are the same as the ones in the other sections of the paper.
> >
> > - top-k sampling
> >   We added the citation (Section 4).
> >
> > [1] Hafez: an Interactive Poetry Generation System, Ghazvininejad’ et. al., ACL’2017
> > [2] Multiple-Attribute Text Rewriting, Lample et. al., ICLR’19
> > [3] What makes a good conversation? How controllable attributes affect human judgments, See et. al., NAACL’19
> > [4]  CTRL: A Conditional Transformer Language Model for Controllable Generation, Keskar et. al., 2019
> > [5] Fine-Tuning Language Models from Human Preferences, Ziegler et. al, 2019
> > [6] Hierarchical Neural Story Generation, Fan et. al., ACL’18
> > [7] Towards controlled text generation,Hu et. al., ICML‘17
> > [8] Controlling Linguistic Style Aspects in Neural Language Generation, Ficler et. al, 2017
> > [9] Style Transfer from Non-Parallel Text by Cross-Alignment, Shen et.al., NeurIPS’17

---

### Official Review · AnonReviewer1 · 2019-10-22
**Official Blind Review #1**

**Rating:** 3

**Review:**

The paper proposes a Plug and Play LM model for controlled natural language generation. Similar to the idea of the Plug and Play Generative Networks for vision, the model plugs in a discriminator, which is either a bag-of-words model or a single layer classifier. The added simple discriminator is then coupled with a pre-trained generative language model such as GPT-2, to obtain a conditional probability for generating controllable text. The authors evaluate the proposed model using human evaluation studies and quantitative perplexity metrics, aiming at measuring the relevance and fluency of the generated text. Their experimental results show that the text generated is fluent and aligned with the desired attributes.

The proposed method is simple and makes sense to me. The idea of how one can make good use of large, pre-trained  generative language models is very neat here. However, I have two main concerns, as follows.

1. The main focuses of the generated text seem to be dramatically changed in an unpredictable way while tailoring the control attributes. In this sense, how useful these kinds of text generation techniques are not clear to me. For example, the first two rows in Table 3 contain two paragraphs with very different main ideas to be conveyed. Similarly for sentences in Table 1. It seems that those sentences talk about very different topics/things to me, although they may reflect the desired control attributes.  Is there an automatic evaluation metric to subjectively evaluate the change of the focuses/ideas of two pieces of text?

2. The model is a straightforward adaption of the Plug and Play Generative Networks from the vision community.

In short, the idea in the paper is simple and seems effective. On the other hand, the lack of a good evaluation metric makes me a bit uncertain about the contribution of the paper. I am willing to increase my evaluation score if I will be convinced by other reviews and comments.


**Experience Assessment:**

I have read many papers in this area.

**Review Assessment: Checking Correctness Of Derivations And Theory:**

I assessed the sensibility of the derivations and theory.

**Review Assessment: Checking Correctness Of Experiments:**

I assessed the sensibility of the experiments.

**Review Assessment: Thoroughness In Paper Reading:**

I read the paper at least twice and used my best judgement in assessing the paper.

---

> ### Author Response · Authors · 2019-11-14
> **Response**
>
> >> The proposed method is simple and makes sense to me... is very neat here. However, I have two main concerns, as follows.
>
> We thank you for your comments helping us improve the paper. We address your comments below.
>
> >> "1. The main focuses of the generated text seem to be dramatically changed in an unpredictable way while tailoring the control attributes. In this sense, how useful these kinds of text generation techniques are not clear to me. .... Is there an automatic evaluation metric to subjectively evaluate the change of the focuses/ideas of two pieces of text?"
>
> This is certainly the case. In our work, two samples from either an LM distribution p(x) or a controlled LM p(x|a) are independent. The task being studied is controlled generation as opposed to style transfer, the latter scenario in which one aims to retain content but adjust style. Our goal is not to control the language so that the idea being conveyed is retained.
>
> Although it would be great if our model could accomplish both feats, we would like to note that controlled generation on its own is an actively studied problem in the language community. Recently several approaches have been proposed towards solving the problem of open-ended controlled generation where the goal is to only generate language with specific attributes without controlling for context, for example, the following papers: [1], [2], [3], [4], [5]. Another paper, [6], showed the benefits of language control (without directly controlling the idea being conveyed) on human judgement of the quality of engagement during interaction with a dialogue agent. We also note that the PPGN model in the paper inspiring this work does not control for deviation in context, but rather only controls for the generated image having the desired attribute (i.e. PPGN and PPLM both perform “controlled generation” but not “style transfer”).
>
> For the open-ended controlled generation task (such as studied in [1,2,3,4]), we consider several possible automatic and human evaluation metrics, including perplexity, dist scores, human fluency and attribute relevance scores. If you have any suggestions for other automatic evaluation metrics, we would be happy to consider including them.
>
> >> "2. The model is a straightforward adaptation of the Plug and Play Generative Networks from the vision community."
>
> We respectfully disagree that the adaptation was straightforward. While we would have been happy to apply the PPGN approach directly to the language domain, the adaptation actually required several modifications, summarized as follows:
> PPGN:
> -- A graphical model depiction of the network looks like this: h -> x -> y, where h is a latent code, x is an image, and y is a class or attribute.
> -- A single h generates an entire, single image x.
> h and x are both continuous, and the gradient w.r.t. y passes through x to h.
> -- The Markov chain is run in h space, with a separate p(h) model being trained and used to ensure h does not drift too far from high probability regions.
> -- Multiple steps are taken in h space, corresponding to multiple entire images.
> -- Noise is added in h space to obtain the correct diversity of images.
> PPLM:
> -- A graphical model depiction of the network looks like this: [x1 -> (h1, x2) -> (h2, x3), … ] -> y, h_t and x_t are the latents and byte-pairs at time t and y is an attribute.
> -- A single h generates a distribution over sentences x.
> h is continuous and x is discrete, and gradient w.r.t. y passes directly to h, with discrete x skipped, except in the distribution propagation approach in Sec 4.3, which propagates through the single word x_t+1 (“Instead, as in -- Dai et al. (2019a), we use the distribution…”).
> -- A complete Markov chain is not run, as this would require multiple full forward and backward passes through the transformer. Instead, we update only a sliding window of the recent past of h and sample only one word at a time. This is a compromise between speed and quality of the samples. The particular dependency structure of the transformer allows us to update only the past (key, value) pairs, which also allows for efficient sampling.
> -- Multiple steps are taken in h space as the sentence is constructed word by word. Multiple entire sentences are never produced.
> -- Noise is added via the sampling of each word in x space to obtain the correct diversity of sentences.
>
> [1] CTRL: A Conditional Transformer Language Model for Controllable Generation, Keskar et al., https://arxiv.org/abs/1909.05858
> [2] Fine-Tuning Language Models from Human Preferences, Ziegler et al., https://arxiv.org/abs/1909.08593
> [3] Towards controlled text generation, Hu et al., https://arxiv.org/abs/1703.00955
> [4] Controlling Linguistic Style Aspects in Neural Language Generation, Ficler et al.,  https://arxiv.org/abs/1707.02633
> [5] Towards Controllable Story Generation, Peng et al.
> [6] What makes a good conversation? How controllable attributes affect human judgments,See et al., NAACL’19

---

> > ### Comment · AnonReviewer1 · 2019-11-15
> > **Thank you for your rebuttal**
> >
> > Thank you very much for your feedback on my reviews; really appreciate that.
> >
> > Regarding "If you have any suggestions for other automatic evaluation metrics, we would be happy to consider including them.", unfortunately, I don't have them, but I do think that measuring the content shifting among the generated texts could be useful. In this sense, we would be able to control the consistency of the targeted context or conversation.

---

> > > ### Author Response · Authors · 2019-11-15
> > > **Thanks for the quick response!**
> > >
> > > Thanks for your quick response!
> > >
> > > We agree with you that controlling for consistency in context is very interesting and in general, the lack of metrics for measuring content shift.
> > >
> > > The current objective of this piece of work is controlled generation as studied in [1,2, 3, 4], as opposed to style transfer where retaining content is more of a concern. We do hope to extend PPLM into style transfer in the future, and also to applications such as NMT ( e.g. this could allow transforming a German to English translation model and a Twitter vs. Wikipedia classifier to translate German phrases into their English Twitter equivalents) where retaining content is extremely important.
> > >
> > > However, in the context of this paper, the factors we are evaluating for 1) can we steer text generation towards desired attributes, and 2) can we do this without degrading language fluency and diversity. We have used 2 types of human annotation (with thousands of labels collected), 4 kinds of automated measures, and evaluations from an additional, separately trained classifier. We believe this presents the most comprehensive evaluation we have seen so far in the literature for the task of controlled generation. We have also included controlled generation baselines as mentioned in our general response.

---

### Official Review · AnonReviewer3 · 2019-10-23
**Official Blind Review #3**

**Rating:** 6

**Review:**

The paper introduces an approach to the conditional generation of text, relying on pre-trained decoders, without fine-tuning and, in certain cases, without any training at all. The approach they introduce is following the framework known in NLP as noisy-channel modeling, previously standard in machine translation (in its SMT days), but undergoing certain revival recently (https://arxiv.org/abs/1611.02554, https://arxiv.org/abs/1910.00553,https://arxiv.org/abs/1908.05731,https://arxiv.org/abs/1907.06616). The authors do not mention this connection (they should!).
Very differently from these previous approaches attempting to integrate the two factors in the search process (e.g., using reranking), the authors instead rely on gradient descent in the latent space of their model (Transformer), similarly to plug-n-play generative networks in image generation.

I find this approach interesting and like the paper overall. However, I do not see why authors do not compare to more direct ways of integrating the conditional component into the model. This would have been tricky in the NMT papers mentioned above, as the entire source sentences need to be reconstructred, however, it should be quite straightforward in this work, with conditioning on single categorical control variables (or maybe a couple in the additional experiments in sect 4.4). Especially, given that the authors already make the predictions of the control variable independently per prediction (e.g., see eq. (5) in section 4.2) / greedily per prefix (bottom lines, page 7). I would actually expect the proposed approach to work better (or at least differently) but it would be interesting to see it confirmed. E.g., for the experiments defining topics as sets of seed words (section 4.2), when integrating factors directly (unlike the proposed approach, Table 3), there will be no increase in the probability of generating relevant words before the first seed word is generated.

Another limitation is the lack of comparison to standard controlled generation work, i.e. those requiring training a model or/and fine-tuning pretrained decoder. I understand that the proposed approach falls in a different category and, of course, do not expect it to beat a fine-tuned model, but I'd like to get some feel for how much one loses by using this simpler method. There has been a lot of work on controlled generation in recent ~3 years, and they can also be combined with intializing and fine-tuning off-the-shelf pretrained decoders.

There is an interesting relation to the NIPS 2019 paper: https://arxiv.org/abs/1907.04944  They also rely on gradient descent to steer a pretrained language model. Their goal is to assess the degree of 'steerability' rather than building a controlled-generation model.

Given that style-controlled but otherwise unconditional generation may not have that many applications, I am curious how far you can push this approach. E.g., can you make it scale to more complicated data-to-text generation tasks (https://www.aclweb.org/anthology/D17-1239/)? Or, will the only application in this context be integrating new conditioning variables into pretrained conditional LMs?

Minor: I am confused with the notation in "Post-norm Geometric Mean Fusion" section.  It says that softmax is applied to the product of probabilities. Maybe to a linear interpolation of log-probs? Or maybe that's not softmax at all? Something seems off here.


**Experience Assessment:**

I have read many papers in this area.

**Review Assessment: Checking Correctness Of Derivations And Theory:**

I carefully checked the derivations and theory.

**Review Assessment: Checking Correctness Of Experiments:**

I carefully checked the experiments.

**Review Assessment: Thoroughness In Paper Reading:**

I read the paper at least twice and used my best judgement in assessing the paper.

---

> ### Author Response · Authors · 2019-11-14
> **Response to review**
>
> Thank you for your comments helping us improve the paper!
>
> >> “following the framework known in NLP as noisy-channel modeling …. this connection (they should!).”
>
> Thanks for pointing out references to noisy-channel modeling. We have added a mention of the noisy channel modeling approach to our related work section and have discussed how that approach compares to PPLM (Section 2).
>
> >> “I find this approach interesting and like the paper overall.”
> Thanks!
>
> >> “However … do not compare to more direct ways of integrating the conditional ... expect the proposed approach to work better (or at least differently) but it would be interesting to see it confirmed. ....  will be no increase in the probability of generating relevant words before the first seed word is generated.”
>
> Thank you for these great suggestions. We’ve updated the paper to include this approach both for PPLM-BoW and PPLM-Discrim models:
> For PPLM-BoW: this corresponds to an existing approach referred to in literature as “Weighted Decoding” (https://www.aclweb.org/anthology/P17-4008/, See et al., NAACL’19).
> For PPLM-Discrim: for each token in the vocabulary we compute p(y=desired sentiment | x) and sample from the distribution p(x)*p(y=sentiment|x) with top-k=5. While the forward passes over the vocabulary are extremely expensive (e.g. 50000x), we get a sense of how well PPLM compares with a direct integration of the condition. We have included both sets of results in the paper (Table 3 and Table 6, row “WD”), where we find, as you presumed, that it does not work quite as well as PPLM.
>
> PPLM works better than directly integrating the conditioning into the decoding procedure. For the bag of words, we can further confirm the observation that the probability of generating relevant words before the first seed word from the bag does not increase. Another key difference is that the semantics of the bag of words are not captured, rather the decoder chooses to pick one of the words that fits context. For instance, a generated sample when conditioned on the prefix “Once upon a time” with the “Space” bag of words is “I used to have a pretty good idea what a starfish was. I was a starfish biologist.”.  See Section 4.2 for details.
>
> For the sentiment control task, we find that this results in a lot of adversarial samples. Sequences often have a high attribute likelihood under the discriminator used during decoding but do not possess the attribute under human evaluation/external classifier evaluation.
>
> >> “Another limitation is the lack of comparison to standard controlled generation.... fine-tuning off-the-shelf pretrained decoders.”
>
> Thanks for the suggestion! We have updated the paper to include comparisons with a recent conditional LM (CTRL, https://arxiv.org/abs/1909.05858) and a GPT-2 LM fine-tuned for positivity with RL and human preferences (https://arxiv.org/abs/1909.08593). The details of the set-up can be found in the Section S7, particuarly, S7.1 and S7.2. We find PPLM performs comparably with CTRL on sentiment control (Table 6) and (perhaps surprisingly) outperforms CTRL on topic control (Table 3). PPLM also significantly outperforms the fine-tuned GPT-2 model on the sentiment task. In all of the above cases, PPLM is at least as fluent or more fluent than the baselines (CTRL & fine-tuned GPT-2). This is impressive considering that the fine-tuned GPT-2 model has over twice as many parameters, and the CTRL conditional language model has over 4 times as many parameters and are specifically tuned/trained for controlled gen.
>
> >>”There .. interesting relation to the NIPS 2019 paper … 'steerability' rather ... controlled-generation model.”
>
> Thanks for the interesting connection! We have included this (Section 2, Page 3).
>
> >> “Given that style-controlled … can push this approach ... pretrained conditional LMs?”
>
> We believe the PPLM approach should scale well to any method with a Transformer based decoder, including potentially the application you describe, or NMT or Dialogue systems, where See et al.’19 showed the utility of being able to control the response in dialogue systems. Just as PPLM allows one to combine a p(x) model and p(a|x) model to generate samples from p(x|a), in the NMT scenario one could combine a base translation model p(x_target | x_source) with a p(a | x_target, x_source) to generate samples from p(x_target | a, x_source). E.g. this could allow transforming a German to English translation model and a Twitter vs. Wikipedia classifier to translate German phrases into their English Twitter equivalents!
>
> These are some of the immediate next steps we plan on exploring!
>
> >>”Minor: ...  Something seems off here.
>
> Indeed this was an error -- we had posted a comment mentioning a correction on openreview. Thanks for pointing out; we have fixed this in the revision now.

---

### Public Comment · ~Jason_Brett1 · 2019-09-27
**Neat tricks but may have some critical defects**

The idea behind this paper is simple but interesting. However, I personally have some questions about this work and hope to get replies from the authors.

(1) This paper does not provide any comparison with other works. For example, for a publicly available conditional language model, CTRL is only mentioned in related work but not compared as a baseline. As a researcher myself, I completely understand that CTRL is recently released and the authors do not have sufficient time to run CTRL. But I wish to see the results as supplemental material later on OpenReview. On the other hand, there are already several conditional generation model, e.g., S-VAE, CTRL-GEN. The authors should at least compare with these works.

(2) The comparison with the vanilla GPT-2 seems to be unreasonable. The GPT-2 baseline is not provided with any condition-specific information, which surely cannot generate conditional text. For example, if providing the GPT-2 with the CTRL-style prompt (e.g., Rating 5.0/5.0, or Topic: Military), I'd like to see how GPT-2 performs with this setting.

(3) To what end is the generated text conditional? The results in Table 4 seem to be significantly lower than prior methods (e.g., CTRL-GEN).

(4) Will the fluency degenerate along with the increase of conditionality? How to balance that? I didn't find an answer in the article.

(5) The BoW strategy seems to be confusing. If we simply multiply the probability of the keywords by a constant larger than one, will it also do the trick? For example, we manually multiply the probability of {best, excellent, wonderful, ...} by 2, can GPT-2 generate positive samples as well? If we measure the cosine similarity between the pretrained word embeddings of predicted words and the given keywords, can we even get better results? (e.g., it will decrease the probability of words like "terrible", "infamous" and increase the probability of "terrific", "gorgeous"). It seems like the work in this paper is an over-sophisticated version of what I just mentioned.

(6) A question about the setting: I wonder whether the setting for sentiment-conditioned generation is too elaborate? Why choose the SST-5 dataset and only use "Very Positive" and "Very Negative" instead of SST-2 dataset with a binary classification or SST-5 with a three-fold classification (i.e., Positive, Neutral, Negative)? The setting in the paper clearly exaggerates the real performance of the proposed method.

Nitpick:
The Dist-1 metric in Table S4 is labeled as smaller better, which is in conflict with Table 4.

---

> ### Public Comment · ~Karl_William_McMara1 · 2019-09-27
> **Great paper with really smart dynamic conditioning**
>
> This paper is really smart in how it performs conditioning of the generated text.
> Unlike conditional generation and finetuning, which is a huge win, as the authors explain in the text, this method allows plugging any classifier and any bag of words as a way to condition the text, and also to change the intensity and the type of the conditioning along the way and also add additional conditioning aspects after the original LM model is trained, something that conditional methods cannot do.
> For this reason I believe this work is in a different class with respect to other works and some of your comments are unfair:
>
> 1) CTRL was released publically less than a week before the ICLR deadline, which means it was entirely impossible for the authors to add anything about it other than citing it. Moreover, because this PPLM approach can do things that CTRL (and other conditional models) can't do and doesn't require any training, they are on different planes. Comparing them would be like comparing an unsupervised model with a supervised model. In the CTRL paper, moreover, they don't compare with anything neither they run a user study, while in this paper there's both a user study and an automatic evaluation, that together make results really convincing.
>
> 2) the comparison with vanilla looks to me like a.way to provide a baseline for fluency, and it's clearly a useful one as the PPLM slightly decreases the fluency in the human evaluation, which is something the reader would want to know. Moreover the authors also have the BC and BR ablations to compare against, which makes the results totally fair.
>
> 3) the extent of the conditioning is clearly shown in table 1, 3, 5, 7 and in the appendix S2, S5 and S6. I guess you are referring to table 4 for the first column, the topical one, where PPLM performs better than vanilla (obviously) and the ablations. In that case, the CTRL paper does not report any such human evaluation numbers, so I'm not sure how can you say from those numbers they they seem significantly lower (lower to what?).
>
> 4) in the article they both talk about a couple hyperparameters ( the amount of gradient and the number of gradient updates) and they report some parameters that work well in practice, plus they also show in S2 and S5 how much both parameters influence the generation topicality and the fluency degeneration.
>
> 5) they KL approach is more principled than the hack you are proposing and could be expanded in the future to non bag of words scenarios, like topic obtained from LDA or other methods that return distributions of words per each topic. I wouldn't call their oversophisticated, I would call yours oversimplified.
>
> 6) I guess that was intended to show the strength of the conditioning. I don't believe it exaggerates the real performance of the model at all, it's just a specific choice about what to condition on to obtain a specific result, I don't see anything bad or unfair about it.

---

> > ### Public Comment · ~Jason_Brett1 · 2019-09-27
> > **Some issues to be clear**
> >
> > Hi Karl,
> >
> > Thanks for your attempt to try to explain these to me. I believe you are one of the authors? If so, I think replying in the role of authors may be more appropriate. However, I'd like to point out something in addition to my comment above.
> >
> > First, I never neglect the interesting and smart dynamic conditioning method proposed in the paper. However, as a normal reader instead of a reviewer, I thought it is not mandatory to also compliment the strengths in the paper.
> > Second, I need to say the CTRL (proposed by Salesforce) and CTRL-GEN ( https://arxiv.org/pdf/1703.00955.pdf ) mentioned in my comment are not the same at all. I should have made it more clear. My bad.
> > Then, I'd like to provide some more explanation for my comment and response to your questions.
> >
> > 1) As I already said in my comment, it was completely fine not to provide a comparison with CTRL. I totally understand that. However, what I kindly ask is if the authors can provide the following experiment results on OpenReview, and this is the spirit of an open review, right?
> >
> > 2) Yes! It is a weak but meaningful baseline. I would certainly add this work as a baseline if I have future work on controlled generation.
> >
> > 3) In Table 1 of CTRL-GEN (NOT CTRL!), the result of CTRL-GEN and S-VAE on SST are both higher than the result shown in the second column of Table 6 in this paper. I know there may be some nuance on experimental settings but it cannot result in an absolute 20% performance drop, right? On the other hand, Table 1,3,5,7 and appendix S2, S5 and S6 which you mentioned are all qualitative results (which may be cherrypicked, more or less). However, I was talking about a quantitive result instead.
> >
> > 4) It is indeed a pro of this paper! I may have missed something in the appendix.
> >
> > 5) Yes. I just wonder if the method proposed in this paper is better than my proposed over-simplified version? Also, I'm not suggesting the authors need to compare my naive simplification. It is just proposed for discussion.
> >
> > 6) However, in the CTRL-GEN paper (https://arxiv.org/pdf/1703.00955.pdf , NOT CTRL!), the setting is more reasonable (as I mentioned in the comment, using SST-2 instead of SST-5) and it yields even better performance (0.851) than Table 6 of this paper (0.696). I believe the experiment in this paper is also a binary generation (+, -) instead of a ternary one (+, 0, -). If it is not, please point out. I am not very sure about that.

---

> > > ### Public Comment · ~Karl_William_McMara1 · 2019-09-27
> > > **answering the clarification**
> > >
> > > Hi Jason,
> > >
> > > I'm not an author of the paper, I guess authors answering will still be anonymous and still marked as authors, like previous years.
> > >
> > > I confused CTRL and CTRL-GEN (of which I was not aware of), sorry about that! My fault.
> > >
> > > 1) I still think they are techniques with very different premises, in particular one is trained and the other is not. A comparison, although being a reasonable request, would be akin to comparing the machine translation capabilities in the GPT-2 paper with a fully supervise MT system, not super informative.
> > >
> > > 3) My bad here for confusing the papers. Looking at CTRL-GEN I guess you are referring to Table 1 and Figure 3. If that is the case, I believe the topicality evaluated by human judges and the automatic accuracy obtained by a pretrained classifier are not comparable at all. But I think that type of evaluation could be requested to the authors of this paper, using the same independently trained classifier. Moreover, being their classifier being trained on a different SST, it would be an even more fair comparison than using the same classifier as discriminator too.
> > >
> > > 5) My guess is that a reweighting and renormalization could have a similar effect of increasing the probability of the terms, although I believe if would not increase the probability of related words that are not in the bow, the words highlighted in dark red in the samples. But I agree with you, it would be interesting to see if that is actually the case.
> > >
> > > 6) That accuracy 0.851 automatically computed score is not comparable with the 0.696 obtained from human evaluation. That said, I think most of the discriminator used in this paper are binary, so using a ternary alternative would be interesting too, although I'm not sure it would add a lot to the paper.

---

> > > > ### Public Comment · ~Jason_Brett1 · 2019-09-27
> > > > **Reply #2**
> > > >
> > > > Hi Karl,
> > > >
> > > > Thanks for the discussion. I hope our discussion is beneficial for both the authors and reviewers!
> > > >
> > > > 1) Yes! Of course, CTRL will have much better result, but it does not compromise this paper. We all agree that the method is a weak baseline method instead of a competitive SOTA. However, a fair comparison with CTRL, the possible SOTA will benefit the NLG community. This part is missing in CTRL due to there was no comparable controlled language model. But here comes the chance.
> > > >
> > > > 3) I'd like to kindly remind you that SST-2 and SST-5 are actually the same dataset. They only differ on labels. Thus, I don't think training discriminator on the other SST is fairer. However, I believe you'd also agree that using only "Very positive" and "Very negetive" samples will make both automatic and human evaluation higher than using SST-2 (which only has "Pos" and "Neg"). It is a pity that these two papers use different evaluation, which indeed cannot be compared directly. I also acknowledged that in my second comment, but adding that I do not believe -20% is only because of the different evaluation. On the other hand, if the authors provide comparison under the same settings, I guess it will be more convincing.
> > > >
> > > > 6) My point is if both CTRL-GEN and this paper use binary generation, they should be at least somehow comparable. That's to say, I won't compare a binary result to a ternary one. On the other hand, as I mentioned above, the different evaluation methods are not directly comparable but that's the only thing I can do since I really want to know to what end can GPT-2 be conditional with this proposed method. A direct comparison will surely help answer that. I sincerely hope to see the results provided by the authors.

---

> > > > > ### Public Comment · ~Karl_William_McMara1 · 2019-09-28
> > > > > **Reply**
> > > > >
> > > > > I actually hope reviewers are not influenced by this discussion, as some of the requests in the original comment and considerations in it are incorrect and may mislead judgment, but hopefully the authors can actually draw something useful from this.
> > > > >
> > > > > 1) I believe this paper is the best conditioned language generation we have seen so far that doesn't require retraining or finetuning, both really expensive to perform given the size of the original GPT-2 model, so, in this specific setting, I would consider it SOTA.
> > > > >
> > > > > 2) different annotations of the same inputs make for different datasets. The difference could even be 50%, but if the two numbers are obtained in totally different settings like in this case (human evaluation vs automatic evaluation with a specific classifier trained on a specific dataset) they are absolutely non comparable. You say you don't believe the difference is only because of the evaluation, I'm saying  the human judges have a completely different standard than the automatic system, thus the difference could have been smaller or bigger or reversed in sign, it would have still meant nothing at all.
> > > > >
> > > > > 6) We agree on being curious to see the same automatic evaluation in CTRL-GEN applied here, to actually have a realistic reference. That said, I do believe that the human evaluation provided in the paper is much stronger evidence of the quality of the results than an automatic evaluation would be.

---

> ### Author Response · Authors · 2019-10-02
> **Response to Karl and Jason's discussion: 1/3**
>
> Dear Jason and Karl,
>
> Thank you for your interest in our paper, including posting questions and comments that will help the community better understand and contextualize our work. You both make several valid points, and we thank you both for this engaging discussion. In addition to the points made by Karl (who we are in no way associated with) -- we want to emphasize that our proposed methodology is Plug-and-Play, thus no re-training or fine-tuning of the language model is needed. We address Jason's comments below, although several of them have already been directly addressed by Karl’s responses.
>
> >>(1) This paper does not provide any comparison with other works. For example, for a publicly available conditional language model, CTRL is only mentioned in related work but not compared as a baseline. As a researcher myself, I completely understand that CTRL is recently released and the authors do not have sufficient time to run CTRL. But I wish to see the results as supplemental material later on OpenReview. On the other hand, there are already several conditional generation model, e.g., S-VAE, CTRL-GEN. The authors should at least compare with these works.
>
> Both CTRL and CTRL-GEN train *conditional language models* -- in contrast, we demonstrate controlled generation with a *pre-trained unconditional language model*. We do not believe a meaningful comparison can be made with a ‘plug-and-play’ method and training a conditional LM (as done in S-GAN, CTRL-GEN and CTRL). That said, we could’ve made the distinction more clear in the paper; thanks for the suggestions!
>
> In the limit of infinite annotated data and compute for training, approaches such as CTRL that directly train p(x|a) will outperform our approach. The main objective of our work is to provide a simple inexpensive alternate approach to directly trained conditional models. We’ve added a note to fully clarify this point in the next draft of the paper.
>
>
>
> >>(2) The comparison with the vanilla GPT-2 seems to be unreasonable. The GPT-2 baseline is not provided with any condition-specific information, which surely cannot generate conditional text. For example, if providing the GPT-2 with the CTRL-style prompt (e.g., Rating 5.0/5.0, or Topic: Military), I'd like to see how GPT-2 performs with this setting.
>
> Our main objective with considering unconditioned GPT-2 text (“B”) in the ablation study is to obtain a baseline for human judges. More precisely, it provides information about the bias induced by asking a human if a sentence is positive, or if it is about the military: do humans squint and try to interpret everything in a manner the question suggests? The data shows that they do, and it was important to measure exactly how much, e.g. see the high 13.0% of unconditioned GPT-2 sentences judged to be about any particular topic!
>
> A second key objective is to illustrate that we retain most of the fluency from vanilla GPT-2 generation.
>
> However, your suggestion of an additional baseline is a great one. We quickly tried it out, and preliminary results show that:
> -- Merely adding the “Topic: xyz” prompt doesn't work too well in general. We did obtain some topical relevance for some topics when using short prefixes, but for longer prefixes and more complex topics, we found it not to work well.
>
>
> For example, with the prefix, “The little girl lived in the woods, and her father was a carpenter” adding “Topic: Military” does not reliably generate military related samples. In contrast, PPLM generates military related samples very reliably with the same prefix.
>
>
> -- For sentiment control, we found that simply adding “Rating x/y” does not work reliably. Whether we added a “Rating 1.0/5.0” or “Rating 0.0/5.0” or “Rating 5.0/5.0” prompt before the actual prefix, we found that this always biased GPT-2 to towards producing positive passages often.
>
>
> -- For more complex tasks such as detoxification, it might not be simple to find an easy “extra prefix” that works.
> We’ve added running a more rigorous evaluation of this approach to our TODO list and hope to complete it and add the results to the paper before the end of the review period. Thanks very much for the idea.
>
> Finally, we also note that the two approaches are complementary -- our conditioning can simply be used as an add on to other forms of condition (such as used in the CTRL paper).

---

> > ### Author Response · Authors · 2019-10-02
> > **Response to Karl and Jason's comments: 2/3**
> >
> > >>(3) To what end is the generated text conditional? The results in Table 4 seem to be significantly lower than prior methods (e.g., CTRL-GEN).
> >
> > Both human evaluation and evaluation via automated methods can be very useful, but unfortunately the percentages reported in each case are not comparable for several reasons:
> > a) Human judgements are often quite different from automated measurements of an attribute. For example, a recent work on controlled generation in the style transfer setting MTR [2] is able to achieve 85%+ with automated evaluation (Table 4, [2]) but in the human evaluation setting the number drops to 69.6% (Table 5, [2]). Further, in DAR [1] -- which also focuses on style transfer -- the human evaluation accuracy is 64% (Table 5, [2])  while the automated evaluation accuracy for DAR in [1]  is over 90%. We hope this is sufficient evidence that automated evaluation and human evaluation numbers are not directly comparable. Note that both DAR/MTR are more recent works that CTRL-GEN.
> >
> > Human judges (both in our paper and [2]) are given the option of saying a sentence is neutral: neither positive/negative. Evaluation with an SST-2 discriminator will classify neutral sentences as positive or negative (binary classifier), resulting in an increased count for both positive/negative accuracies. This in turn results in exaggerated numbers for sentiment accuracy. Please note that human annotation in our paper (as in [2]) is not “binary” but rather “ternary”, as the human can mark if a sentence is positive or negative or neutral (if the human marks a sentence as not positive and not negative).
> >
> > b) Additionally, CTRL-GEN evaluated sequences of length<15. In contrast, our numbers are reported on sequences of length 80 (for topic) and sequences of length 50 (for sentiment). This difference further prohibits direct comparison.
> >
> >
> > c) While, the premise for PPLM is entirely different from DAR/MTR -- PPLM is generation from an unconditional model, note that we perform comparably to the human evaluation numbers reported in [2] for both DAR/MTR (more recent works than CTRL-GEN). Further, [2] reports the perplexity for CTRL-GEN (Hu et al., 2017) as 232.0 (significantly higher than other competing approaches).  Also, note that conditioning can be increased at the cost of fluency (See Table S5, Supplementary Information) -- in that setting the topic/sentiment accuracy numbers would go up at the cost of fluency.  In this regard, it makes sense to compare both accuracy and fluency between different approaches as done in [2].
> >
> >
> >
> >
> > >>(4) Will the fluency degenerate along with the conditionality? How to balance that? I didn't find an answer in the article.
> >
> > Yes, fluency does degenerate with the strength of condition as shown in the paper. You can control that by decreasing the step-size alpha, increasing the coefficient for the kl-loss and also decreasing lambda_gm  (in the limit, as lambda_gm goes to 0, you recover the original LM). We found a good set of hyper-parameters that work well in practice (Table S1 in Supplementary Information). We will make this clearer during revision.
> >
> >
> > >>(5) The BoW strategy seems to be confusing. If we simply multiply the probability of the keywords by a constant larger than one, will it also do the trick? For example, we manually multiply the probability of {best, excellent, wonderful, ...} by 2, can GPT-2 generate positive samples as well? If we measure the cosine similarity between the pre-trained word embeddings of predicted words and the given keywords, can we even get better results? (e.g., it will decrease the probability of words like "terrible", "infamous" and increase the probability of "terrific", "gorgeous"). It seems like the work in this paper is an over-sophisticated version of what I just mentioned.
> >
> > The approach you describe with increasing the probabilities of the bag-of-word tokens (referenced to in prior work as ‘weighted decoding’ [3]) will work to some extent but will not result in samples with subtle changes of topics that result from promoting *related* words to the bag of words but not necessarily words directly from the bag. We provide more discussion in extended related work about weighted decoding [3], in S1 in Supplementary Information. For example, consider the sentences generated with topics ‘Politics’, ‘Science’, ‘Fantasy’ in Table 3, and topic ‘Fantasy’  in Table 7 -- here words closely related to the topic appear before words from the bag.

---

> > > ### Author Response · Authors · 2019-10-02
> > > **Response to Karl and Jason's discussion 3/3**
> > >
> > > >>(6) A question about the setting: I wonder whether the setting for sentiment-conditioned generation is too elaborate? Why choose the SST-5 dataset and only use "Very Positive" and "Very Negative" instead of SST-2 dataset with a binary classification or SST-5 with a three-fold classification? The setting in the paper clearly exaggerates the real performance of the proposed method.
> > >
> > > It was a simple design choice to use SST-5 over SST-2. We do not measure with SST-2 (or SST-5) and we use humans and an external dataset (IMDB reviews) to measure sentiment controllability. In this regard, the results are meaningful and fair.
> > >
> > > >>Nitpick:
> > > The Dist-1 metric in Table S4 is labeled as smaller better, which is in conflict with Table 4.
> > >
> > > Thanks for the catch on the Dist-1 “lower is better” typo! We have corrected this and it will appear in the next posted draft.
> > >
> > >
> > > We once again thank you both for your interest, and we’d be happy to continue the discussion.
> > >
> > > [1] Delete, Retrieve, Generate: A Simple Approach to Sentiment and Style Transfer
> > > https://arxiv.org/abs/1804.06437
> > > [2] Multiple-Attribute Text Rewriting, https://openreview.net/forum?id=H1g2NhC5KQ
> > > [3] What makes a good conversation? How controllable attributes affect human judgements. https://www.aclweb.org/anthology/N19-1170/
> > > [4] http://www.abigailsee.com/2019/08/13/what-makes-a-good-conversation.html;

---

### Public Comment · ~Eric_Wallace1 · 2019-09-30
**Any Estimated Time of Code Release?**

Hi, thanks for the paper! I was interested to take a closer look at the implementation of your method. I see you are working on getting approval to release your code, do you have any estimate on when it will be available? If its a while (I know getting company approval can be slow/annoying), is it possible to share it privately?

---

> ### Author Response · Authors · 2019-10-02
> **Code Release**
>
> Hi Eric,
>
> Thanks for the interest in our paper! We are actively working on institute approval, we hope to have it available in the next few days (~ 1-2 weeks). We will keep you updated about this!

---

> > ### Author Response · Authors · 2019-10-10
> > **Code available**
> >
> > Hi Eric,
> > The code is now in the dropbox folder. You should be able to check-out the implementation.

---

### Public Comment · ~Jason_Brett1 · 2019-10-04
**Reply to Authors**

Thanks for your reply! The response is sound and makes sense in every way. I do hope to see this paper get accepted.

Good luck,
Jason

---

### Author Response · Authors · 2019-10-10
**Minor Corrections**

We would like to point the readers to 2 minor corrections in the paper -- we will fix this during revision:
1. Instead of the Softmax normalization in Section 3.3, page 5, paragraph 3, we actually divide by a normalizing factor such that it forms a valid distribution.
2. Table S1, Row corresponding to POSITIVE, NEGATIVE --> gamma_gm = 0.9 is incorrect, and should be gamma_gm = 0.95.

---

### Author Response · Authors · 2019-11-15
**General Response**


Dear Reviewers,

Thank you for your comments helping us improve the paper! We appreciate the time/effort you all have taken in reviewing our paper carefully and giving insightful feedback. We have updated a great deal of our paper thanks to your feedback and to address the concerns raised. To summarize, the main changes are:
a) Added strong baselines: CTRL (a conditional language model with 1.6B parameters; https://arxiv.org/abs/1909.05858), and GPT2-FT-RL (a pre-trained 774M GPT-2 model fine-tuned for positivity; https://arxiv.org/abs/1909.08593), as well as Weighted Decoding, a more direct conditioning approach suggested by Reviewer #3.
b) Performed full evaluations on all those baselines, including over 2000 additional human annotations.
c) Found that PPLM outperforms GPT2-FT-RL and performs comparably with CTRL, even though both of them are trained specifically for conditioned generation, and are over 4 times and twice of our model size (and 5 orders of magnitude larger than our attribute models). Further, PPLM outperforms Weighted Decoding significantly on both topic and sentiment control.
d) Extended related work section where neural style transfer, weighted decoding, and most recent work on controlled generation are included. We thank you for all the suggested references!
e) Extended analysis for settings in which PPLM succeeds and fails. Thanks to Reviewer #2 for the suggestions!

---

### Public Comment · ~Zhiyu_Lin1 · 2020-02-11
**What is the difference of this method vs. back-propagation?**

I'm wondering since K and V of the transformer network is the target and "shift the history H_t in the direction of the sum of two gradients", what is the difference between actually back-propagating this gradient in the backward phase (step 2 in the illustration) and the method described? By backprop into the model, isn't that fine-tuning?

---

> ### Author Response · Authors · 2020-02-11
> **PPLM vs Fine-tuning**
>
> Hi,
>
> Thank you for the insightful questions. We clarify below.
>
> 1. K and V are activations not weights of the model. Usually, with backprop you compute the gradient with respect to the weights of the model because you want to update. We don’t update the weights but we update the activations, which are dynamically determined at each step by encoding the input.
>
> 2. Fine-tuning implies having a pre-trained model that you update. Here the pre-trained LM is untouched, and the updated components are initialized at 0 (no pre-training).
>
> 3. For conventional fine-tuning, you increase the likelihood of a given set of sequences by updating your model weights. PPLM does not update the model or increase the likelihood of a set of sequences directly.

---

### Decision · Program_Chairs · 2019-12-19

**Decision:**

Accept (Poster)

**Comment:**

This paper proposes a simple plug-and-play language model approach to the problem of controlled language generation. The problem is important and timely, and the approach is simple yet effective. Reviewers had some discussions whether  1) there is enough novelty, 2) evaluation task really shows effectiveness, and 3) this paper will inspire future research directions.

After discussions of the above points, reviewers are leaning more positive, and I reflect their positive sentiment by recommending it to be accepted. I look forward to seeing this work presented at ICLR.